# A versatile reverse genetics platform for SARS-CoV-2 and other positive-strand RNA viruses

Alberto A. Amarilla[1,9], Julian D. J. Sng[1,9], Rhys Parry [1,9], Joshua M. Deerain[2,9], James R. Potter[1,9], Yin Xiang Setoh [1,3,9], Daniel J. Rawle [4], Thuy T. Le[4], Naphak Modhiran [1], Xiaohui Wang[1], Nias Y. G. Peng [1], Francisco J. Torres [1], Alyssa Pyke [5], Jessica J. Harrison[1], Morgan E. Freney[1], Benjamin Liang [1], Christopher L. D. McMillan [1], Stacey T. M. Cheung[1], Darwin J. Da Costa Guevara[1], Joshua M. Hardy [6], Mark Bettington[7], David A. Muller[1], Fasséli Coulibaly [6], Frederick Moore[5], Roy A. Hall[1,8], Paul R. Young[1,8], Jason M. Mackenzie [2,10✉], Jody Hobson-Peters [1,8,10✉], Andreas Suhrbier [4,8,10✉], Daniel Watterson [1,8,10✉] & Alexander A. Khromykh [1,8,10✉]

The current COVID-19 pandemic is caused by the severe acute respiratory syndrome coronavirus 2 (SARS-CoV-2). We demonstrate that despite the large size of the viral RNA genome (~30 kb), infectious full-length cDNA is readily assembled in vitro by a circular polymerase extension reaction (CPER) methodology without the need for technically demanding intermediate steps. Overlapping cDNA fragments are generated from viral RNA and assembled together with a linker fragment containing CMV promoter into a circular full-length viral cDNA in a single reaction. Transfection of the circular cDNA into mammalian cells results in the recovery of infectious SARS-CoV-2 virus that exhibits properties comparable to the parental virus in vitro and in vivo. CPER is also used to generate insect-specific Casuarina virus with ~20 kb genome and the human pathogens Ross River virus (Alphavirus) and Norovirus (Calicivirus), with the latter from a clinical sample. Additionally, reporter and mutant viruses are generated and employed to study virus replication and virus-receptor interactions.

[1] School of Chemistry and Molecular Biosciences, University of Queensland, St Lucia, QLD, Australia. [2] Department of Microbiology and Immunology, Peter Doherty Institute for Infection and Immunity, University of Melbourne, Melbourne, VIC, Australia. [3] Microbiology and Molecular Epidemiology Division, Environmental Health Institute, National Environmental Agency, Singapore, Singapore. [4] QIMR Berghofer Medical Research Institute, Herston, QLD, Australia. [5] Queensland Health Forensic & Scientific Services, Queensland Department of Health, Coopers Plains, QLD, Australia. [6] Infection & Immunity Program, Biomedicine Discovery Institute and Department of Biochemistry and Molecular Biology, Monash University, Clayton, VIC, Australia. [7] School of Medicine, University of Queensland, Kelvin Grove, QLD, Australia. [8] Australian Infectious Diseases Research Centre, Global Virus Network Centre of Excellence, Brisbane, QLD, Australia. [9]These authors contributed equally: Alberto A. Amarilla, Julian D. J. Sng, Rhys Parry, Joshua M. Deerain, James R. Potter, Yin Xiang Setoh. [10]These authors jointly supervised this work: Jason M. Mackenzie, Jody Hobson-Peters, Andreas Suhrbier, Daniel Watterson, Alexander A. Khromykh. ✉email: jason.mackenzie@unimelb.edu.au; j.peters2@uq.edu.au; andreas.suhrbier@qimrberghofer.edu.au; d.watterson@uq.edu.au; a.khromykh@uq.edu.au

Positive-strand RNA viruses encompass a large number of viruses from 41 virus families that are assigned to four large orders[1] and include major human pathogens such as the flaviviruses, e.g., West Nile virus and Zika virus (ZIKV); alphaviruses, e.g., chikungunya virus (CHIKV) and Ross River virus (RRV); picornaviruses, e.g., poliovirus and the caliciviruses, e.g., human norovirus (HuNoV); and the pathogenic coronaviruses, e.g., severe acute respiratory syndrome coronavirus (SARS-CoV), Middle East respiratory syndrome coronavirus (MERS-CoV), and the most recent SARS coronavirus 2 (SARS-CoV-2). Traditionally, reverse genetic systems for positive-strand RNA viruses are based on assembling full-length cDNA clones in a plasmid vector or, for larger viruses like coronaviruses, in bacterial or yeast artificial chromosome vectors (BAC and YAC, respectively) and propagating them in bacteria or yeast[2,3]. Incorporating bacteriophage T7 or SP6 promoters upstream of the viral 5′UTR sequence allows cell-free generation of viral RNA with T7 or SP6 RNA polymerases. The incorporation of eukaryotic expression promoters instead of T7 or SP6 promoters enables the generation of viral RNA from transfected DNA by host cell RNA polymerase II[2,4]. While these reverse genetics approaches have been extensively used in the RNA virology field, propagation of full-length cDNA clones, particularly for larger viruses, is problematic due to toxicity of some viral sequences for bacteria and/or yeast and the presence of cryptic transcription, splicing, and termination signals, which often leads to deleterious mutations and/or deletions[3]. Several approaches have been developed to overcome these issues, including (i) the use of very low copy number plasmids, (ii) mutating cryptic sites, (iii) generating full-length DNA templates for in vitro RNA transcription by in vitro ligation of DNA fragments, and (iv) cotransfecting a mixture of overlapping DNA fragments with the first fragment containing eukaryotic expression promoter upstream of the viral 5′UTR sequence[3,5]. While proven to be useful for some positive-strand RNA viruses, these approaches have either not been successful or have not been attempted for most RNA viruses. These approaches also often require bespoke optimization of conditions, e.g., use of specialized plasmid vectors and bacterial strains, limited choice of fragments due to specific locations of restriction sites, and the requirement for large sequence overlaps.

Herein, we advanced the circular polymerase extension reaction (CPER) methodology that we previously developed for flaviviruses[6–8] to allow the generation of RNA viruses that have large genomes and that contain polyA tails such as SARS-CoV-2 (~30 kb) and the insect-specific member of the same viral order, Casuarina virus (~20 kb). We also successfully applied this CPER method to polyA tail-containing representatives of other positive-strand RNA virus families, such as the arthritogenic alphavirus RRV and two caliciviruses, murine norovirus (MNV) and HuNoV, with the latter generated from a human fecal sample. We further demonstrate the utility of this CPER methodology by generating reporter and mutant viruses and illustrate their application in the studies of virus replication and virus–receptor interactions. The ability to rapidly generate and manipulate RNA viruses using this CPER approach provides a robust avenue to facilitate fundamental discoveries[9,10] and ultimately help develop new interventions[11].

## Results

**Developing CPER for SARS-CoV-2.** Coronaviruses are positive-strand RNA viruses with some of the largest RNA virus genomes ranging from 26.4 to 31.7 kb. They belong to the *Coronaviridae* virus family in the order of *Nidovirales*[12]. The virus family generally causes respiratory tract infections in humans and includes common cold coronaviruses, MERS, SARS-CoV, and SARS-CoV-2. The

disease caused by SARS-CoV-2 is called Coronavirus disease 2019 (COVID-19), where symptoms can include the often fatal acute respiratory distress syndrome. As of March 2021, more than 116 million people have been infected, with more than 2.5 million deaths[13].

The SARS-CoV-2 genome is ~30 kb long, encodes as many as 13 ORFs processed into 26 viral proteins, has short 5′ and 3′ untranslated regions (UTR) and, importantly, has a polyA tail at the end of the 3′UTR (Fig. 1a)[14]. To generate SARS-CoV-2 using CPER, viral RNA from passage 4 (P4) of the Australian SARS-CoV-2 isolate QLD02 (GISAID accession EPI_ISL_407896) was used as a template to generate first-strand cDNA (Fig. 1a). The cDNA was then used to PCR amplify six fragments that collectively encompassed the SARS-CoV-2 genome using high-fidelity DNA polymerase (Fig. 1a, b). Each fragment contained overlapping sequences of only 20 nucleotides, although these were selected for high GC content to allow the use of more optimal annealing temperatures (Fig. 1a). To facilitate DNA circularization and viral RNA transcription in eukaryotic cells, the linker fragment contains the last 20 nucleotides of SARS-CoV-2 3′UTR (that overlap with fragment F6), a polyA tail containing 30 adenines and the hepatitis delta virus ribozyme (HDVr) to generate the authentic 3′end of viral RNA, an SV40 polyA signal for efficient transcription termination, a spacer sequence to separate the functional elements, a CMV promoter for in vivo transcription of viral RNA by the cell RNA polymerase II, and the first 37 nucleotides of SARS-CoV-2 5′UTR (that overlap with fragment F1) (Fig. 1a).

Full-length SARS-CoV-2 cDNA was assembled from the viral cDNA fragments and the linker fragment into a circular DNA in a single CPER reaction using a high-fidelity DNA polymerase. The process does not require any of the intermediate steps commonly used in other coronavirus reverse genetics systems[2,15–17]. The CPER reaction mix (Supplementary Fig. 1), without any further purification, was then directly transfected into HEK293T cells. To recover the virus, transfected HEK293T cells were cocultured with the highly permissive Vero E6 cells. Two independent transfection experiments were performed (CPER1 and CPER2), with both yielding infectious virus. The CPER viruses were amplified once in Vero E6 cells to generate viral stocks. CPER3 virus was generated by transfecting the CPER reaction mix into ACE2-expressing HEK293T cells[18] and amplifying the recovered virus in Vero E6 cells.

Nanopore sequencing of the viral cDNA fragments PCR-amplified from P4 viral cDNA showed they had the same swarm/quasi-species sequence variation as the wild-type (WT) P4 virus (Supplementary Table 1). Comparison of the consensus genome sequences between P4 viral cDNA and PCR-amplified fragments used in CPER assembly (deposited to GenBank, ID: MW772455) did not identify any changes. These data illustrate that the high-fidelity PCR used to generate the cDNA fragments that were subsequently used for the CPER reactions had faithfully amplified the original viral sequence.

Culture of SARS-CoV-2 viruses in Vero E6 cells is well known to select rapidly for mutations, often in or around the furin cleavage site[19–22]. This remains true for a number of reverse genetics systems[16,21,23], although deep sequencing of recovered viruses is not always provided[15,17,24–26]. Such selection was also seen for CPER viruses recovered from Vero E6 cells (cocultured with transfected HEK293T) (Supplementary Table 1). The selection of certain amino acid changes was different for CPER1, 2, and 3 viruses (Supplementary Table 1), with such variability in selection also reported for other reverse genetics systems[16,21,27]. As noted previously[21,22], the choice of cell lines can affect these selection processes, with our recovery of CPER3 virus using ACE2-HEK293T cells showed a decrease in the number of

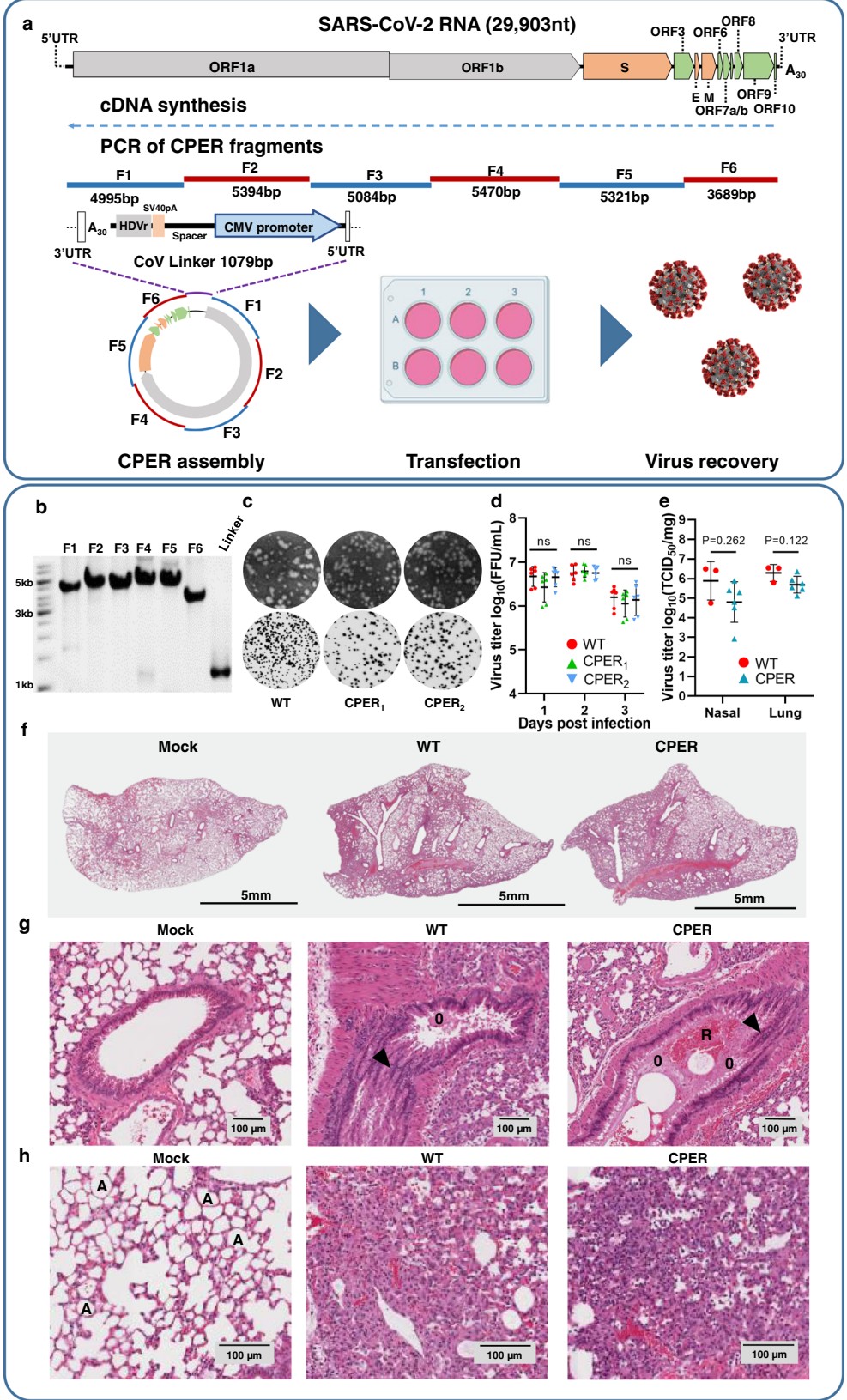

affected sites from 4 to 1 (Supplementary Table 1). Given the SARS-CoV-2 genome encodes >9700 amino acids, the CPER method using ACE2-HEK293T cell thus achieved nearly complete (99.99%) amino acid sequence fidelity.

To examine viral properties in cells, a standard plaque assay with crystal violet staining of infected cells at 2 days post infection (dpi) was performed and showed similar plaque sizes for CPER1 and CPER2 viruses and the parental WT QLD02 isolate (Fig. 1c,

**Fig. 1 Generation of SARS-CoV-2 by CPER and characterization of properties of recovered viruses in cells and mice. a** Schematics of SARS-CoV-2 genome and overlapping SARS-CoV-2 fragments amplified from SARS-CoV-2 cDNA and circularized with a linker fragment containing the last 20 nucleotides of SARS-CoV-2 3'UTR, 30As, hepatitis delta virus ribozyme (HDVr), SV40 pA signal for transcription termination, spacer sequence, CMV promoter, and first 37 nucleotides of SARS-CoV-2 5'UTR. The resultant SARS-CoV-2 CPER product was then directly transfected into HEK293T cells, then cocultured with Vero E6 cells for virus recovery. **b** Agarose gel electrophoresis of PCR-amplified SARS-CoV-2 fragments 1–6 and the linker fragment showing a representative image of three experimental repeats. **c** Representative plaque morphologies of the wild-type (WT) SARS-CoV-2$_{QLD02}$ isolate and CPER-recovered viruses (CPER1 and CPER2) stained using crystal violet at 2 dpi (top) and Immuno-plaque assay (iPA) with anti-spike protein monoclonal antibody at 14 hpi (bottom). **d** Growth kinetics of WT (red) and CPER-generated viruses (green and light blue) over a 3-day time course in Vero E6 cells infected at a multiplicity of infection MOI = 0.01, $n = 2$ independent experiments with three replicates in each, statistical analysis was performed by two-way analysis of variance with Tukey's multiple comparisons test against WT virus. Mean values for each virus at each time point are shown ± SD. **e** End-point virus titers of nasal turbinates and lung tissues from JAX K18-hACE2-transgenic mice infected intranasally with $8 \times 10^4$ FFU/mouse of WT SARS-CoV-2$_{QLD02}$ isolate (red) and CPER-recovered viruses (blue). At 5 days post infection, mice were sacrificed, and virus titers were determined by TCID$_{50}$ assay on Vero E6 cells. For statistical analysis between WT and CPER viruses, unpaired $t$-test with Welch's correction was used, $p$ values are two-sided. Mean values for each treatment are shown ± SD. For WT virus, three biological replicates were used, for CPER-generated virus six biological replicates were used, results are from one experiment. **f** Full lungs from mice infected with WT virus, representative CPER virus, or infected (mock), harvested at 5 days post infection and stained with hematoxylin and eosin (H&E). Scale bar is 5 mm. **g** Selected H&E stained images of lung sections from mice infected with WT or representative CPER virus showing sloughing of the bronchial epithelium as indicated by the arrowhead, bronchi occluded with edema (O) and red blood cells (R). **h** Additional features of SARS-CoV-2 infection in lungs of infected mice showing the collapse of alveolar spaces (A) and edema. The scale bar is 100 μm. Representative images from **f–h** are from three independently analyzed samples for each treatment. Source data for **d** and **e** are provided in the Source Data file.

top row). Immuno-plaque staining (immuno-plaque assay, iPA) with anti-spike protein monoclonal antibody CR3022 showed clear viral foci (Fig. 1c, bottom row). This method was used to determine viral titers (foci forming units, FFU) as it allows detection of viral foci as early as 14 h post infection (hpi). For comparison of virus replication properties in vitro, Vero E6 cells were infected with CPER1, CPER2, and WT QLD02 viruses at MOI = 0.01, with virus titers determined by iPA on 1, 2, and 3 dpi. Both CPER viruses replicated with comparable kinetics to the parental WT QLD02 virus (Fig. 1d).

To compare virus properties in vivo, the JAX K18-hACE2-transgenic mouse model was used. These mice develop a respiratory disease resembling severe COVID-19[28,29], but also present with a fulminant brain infection that is associated with mortality[30]. Although the virus can be found in the brain of ~20% of COVID-19 patients, neurological signs and symptoms are thought to arise from systemic reactions or complications rather than being associated with extensive brain infection[31]. Groups of three JAX K18-hACE2-transgenic mice were infected intranasally with $8 \times 10^4$ FFU/mouse of QLD02, CPER1, and CPER2 viruses. On day 5 after infection, mice were sacrificed, and virus titers in nasal turbinates and lungs were determined by TCID$_{50}$ assays using Vero E6 cells. Viral titers were similar for all three viruses with no significant differences in lung titers or nasal turbinate titers between WT QLD02 ($n = 3$ replicates) and CPER1 and CPER2 viruses (combined $n = 6$ replicates) ($p = 0.12$ and 0.2, respectively, $t$-tests with Welch's correction) (Fig. 1e). Lungs were harvested at 5 dpi and processed for histology, with hematoxylin and eosin (H&E) staining showing a series of profound pathological changes that were similar between CPER-derived and WT viruses (Fig. 1f). These included bronchioles occluded with edema, red blood cells and sloughing of the bronchial epithelium (Fig. 1g), the collapse of alveolar spaces (Fig. 1h), predominantly mononuclear cell infiltrates (Supplementary Fig. 3a), thickening of alveolar septa (Supplementary Fig. 3b), smooth muscle hypertrophy/hyperplasia (Supplementary Fig. 3c), and edema in the extracellular matrix (Supplementary Fig. 3d). Automated quantitation of lung consolidation and cellular infiltration also showed no significant differences between CPER-derived and WT viruses (Supplementary Fig. 3e, f). While the number of animals was small, collectively the data on viral titers in nasal turbinate and lungs, and pathological changes in lungs indicate that CPER-derived and parental WT QLD02 viruses behave similarly in vivo.

**Utilizing CPER for the generation of SARS-CoV-2 D614G mutant and ZsGreen-expressing reporter virus.** To further illustrate the utility of CPER as a SARS-CoV-2 reverse genetic system, we introduced an amino acid mutation, D614G, into the spike protein of the QLD02 isolate. SARS-CoV-2 variants carrying the D614G mutation have become the dominant circulating viruses worldwide[32]. SARS-CoV-2 D614G isolates have recently been shown to produce higher infectious titers in the nasal washes and trachea of infected hamsters, suggesting that the D614G mutation may enhance viral loads in the upper respiratory tract and subsequently could increase transmission[25,26]. To introduce the D614G mutation into the genetic background of QLD02, fragment F5 was split into two overlapping fragments, F5A and F5B, each containing GAT (D) to GGT (G) codon substitution in the overlapping region (Fig. 2a). These two fragments were amplified from QLD02 viral cDNA (Fig. 2b) and incorporated into the CPER assembly instead of fragment F5. The CPER mix contained the remaining fragments F1–F4, F6, and the linker fragment (Fig. 2a, b) and used the same cycling conditions. The mutant virus was recovered by transfecting the CPER mix into HEK293T cells and coculturing them with Vero E6 cells. The D614G virus formed similar size plaques to the parental QLD02 isolate, but both QLD02 and D614G viruses produced slightly larger size plaques than a recent QLD935 viral isolate (Fig. 2c). QLD935 isolate is a more recent viral isolate containing the D614G mutation and other amino acid changes (NSP3 C1392F, NSP3 T835I, NSP12 P323L) (GISAID accession EPI_ISL_436097). The D614G mutation was verified by Sanger sequencing of the RT-PCR-amplified fragment F5 generated from viral RNA (Fig. 2d). Replication efficiencies of D614G and QLD02 in Vero E6 cells were similar, while QLD935 exhibited reduced replication on day 1, but not on day 3 (Fig. 2e). Hence, these results show that the D614G mutation alone is not responsible for the smaller plaque size and delayed replication of the QLD935 isolate, with other changes in the genome responsible for this phenotype.

CPER was also employed to generate a SARS-CoV-2 reporter virus expressing ZsGreen fluorescent protein. The reporter virus was constructed by replacing codons for amino acids 14-108 of ORF7a with ZsGreen sequence, similar to the reporter SARS-CoV-2 viruses generated by other reverse genetics systems[15–17,27]. The CPER methodology for generating ZsGreen reporter virus was carried out using the linker fragment, SARS-CoV-2 fragments F1-F5, two subfragments of fragment F6 (F6A and F6B), and a

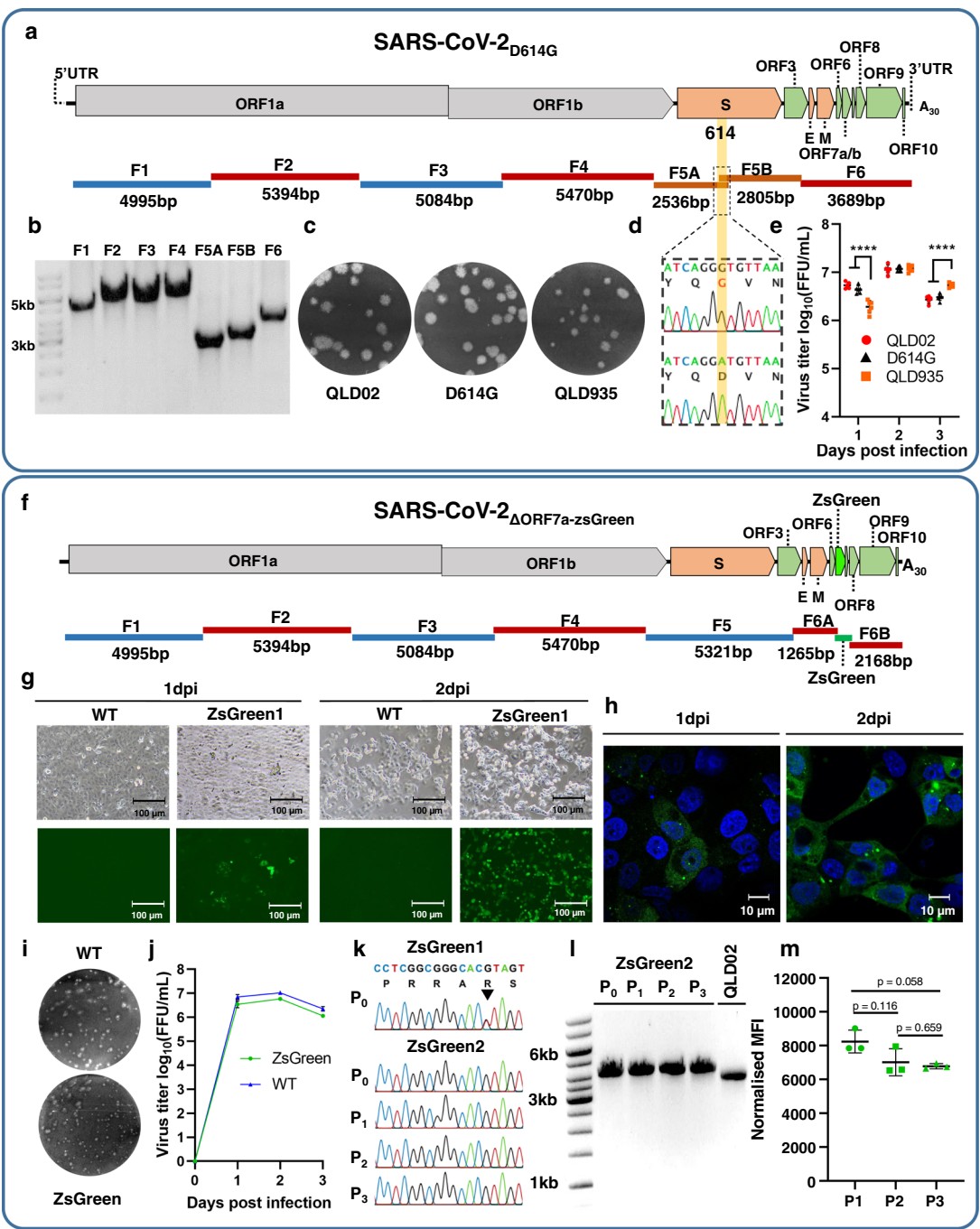

fragment containing ZsGreen with overlapping ends for fragments F6A and F6B (Fig. 2f). Again, no intermediate cloning was required. The reporter virus ZsGreen1 was recovered using the same protocol as for the D614G mutant except that RNA from passage 3 of the QLD02 virus was used to generate viral cDNA for amplifying CPER fragments. In addition, TMPRSS2-expressing Vero E6 cells were generated (Supplementary Note 1) and used in a separate experiment to coculture with CPER-transfected HEK293T cells to generate the ZsGreen2 virus (Supplementary Fig. 2). Nanopore sequencing of the reporter viruses showed significantly less variation in the furin cleavage site compared to CPER1-3 viruses, with the least variation observed in ZsGreen2 virus generated in TMPRSS2-Vero E6 cells (Supplementary Table 2).

ZsGreen expression was detected in Vero E6 cells infected with the reporter virus, but not the WT virus (Fig. 2g). ZsGreen

showed ER localization and formed visible intracellular foci (Fig. 2h), likely because amino acids 1–13 of the signal sequence (at the N-terminus) and the ER retention sequence KRKTE (at the C-terminus) of ORF7a were retained. The reporter virus formed slightly smaller plaques at 2 dpi (Fig. 2i) and showed some reduction in replication in Vero E6 cells while still replicating to high titers of ~$6 \times 10^6$ focus-forming units per milliliter (FFU/mL) (Fig. 2j). To assess reporter virus stability, ZsGreen2 was passaged three times on TMPRSS2-Vero E6 cells, and viral RNA from each passage was subjected to RT-PCR for fragment F6 to assess ZsGreen insertion and for fragment F5 to sequence the furin cleavage site. ZsGreen insertion was stably retained, and furin cleavage site remained unchanged in the viral RNA at all passages (Fig. 2k, l). Flow cytometry of infected cells (Supplementary Fig. 4a, b) also showed that ZsGreen fluorescence

**Fig. 2 Generation by CPER and characterization of SARS-CoV-2 D614G mutant virus and ZsGreen reporter virus. a** Schematics of D614G mutant virus genome and overlapping fragments used to introduce D614G mutation by CPER. Fragment 5 is split into subfragments 5A and 5B with their overlapping region incorporating D614G mutation. **b** Agarose gel electrophoresis of PCR-amplified SARS-CoV-2 fragments representative of at least three experiments. **c** Representative plaque morphologies of wild-type (WT) SARS-CoV-2$_{QLD02}$ isolate, CPER-recovered D614G mutant virus, and SARS-CoV-2$_{QLD935}$ isolate naturally harboring the D614G mutation. Infected cells were stained with crystal violet at 3 dpi. **d** Sanger sequencing of CPER-generated D614G mutant virus (top) and wild-type SARS-CoV-2$_{QLD02}$ cDNA (bottom). **e** Growth kinetics of SARS-CoV-2$_{QLD02}$ isolate (red circle), CPER-generated D614G mutant virus (black triangle), and SARS-CoV-2$_{QLD935}$ (orange square) isolate over a 3-day time course in Vero E6 cells infected at MOI = 0.01, $n = 2$ independent experiments with three replicates in each. Statistical tests to examine differences in growth kinetics between virus isolates were analyzed using a two-way ANOVA. The differences between each time point were analyzed with Tukey's multiple comparisons test. Adjustments to $p$ values were not made for multiple comparisons. ****$p \leq 0.0001$. Bar graph is shown as mean values ± SD. **f** Schematics of SARS-CoV-2$_{\Delta ORF7a-ZsGreen}$ reporter virus and overlapping fragments used to generate this virus. Fragment 6 is split into two subfragments, 6A and 6B to generate a 95 codon deletion in ORF7a (deleted codons 14-108), and ZsGreen gene is inserted in the place of this deletion. Images of ZsGreen fluorescence of Vero E6 cells infected with MOI = 0.1 of CPER-generated reporter virus taken at 40× magnification (**g**) and 100× magnification (**h**). Representative images from **g** and **h** are from three independently analyzed samples from each treatment and two independent experiments. **i** Representative plaque morphologies of wild-type (WT) SARS-CoV-2$_{QLD02}$ isolate and CPER-generated ZsGreen reporter virus stained with crystal violet at 2 dpi. **j** Growth kinetics of SARS-CoV-2$_{QLD02}$ (WT, blue) and CPER-generated reporter virus (ZsGreen, green) over a 3-day time course in Vero E6 cells infected at MOI = 0.01. $n = 2$ independent experiments with three replicates in each. Graph is shown as mean values ± SD. **k** Sanger sequencing showing positions 23603-23620 of the SARS-CoV-2 QLD02 isolate corresponding to the polybasic furin cleavage site (RRAR) in the spike protein. ZsGreen1, amplified in Vero E6 cells, shows ambiguous peak at 23616 (indicated with the arrowhead) corresponding to variable deep sequencing data at this position. ZsGreen2, amplified and passaged on Vero E6-TMPRSS2 cells, shows no change to the furin cleavage site over three passages. **l** RT-PCR with the CoV-6F/6R primer pair of fragment 6 containing ZsGreen insertion compared to the parental QLD02 isolate showing retention of insertion over three passages in Vero E6-TMPRSS2 cells. The ZsGreen2 virus amplifies a 4094 nt fragment, whereas the WT QLD02 virus amplifies a 3689 nt fragment. Gel electrophoresis is representative of at least three experiments. **m** Median fluorescence intensity (MFI) values from flow cytometry analysis (Supplementary Fig. 4) normalized to viral titers (MFI/log$_{10}$(FFU/mL)) of ZsGreen2 virus over three passages in Vero E6-TMPRSS2 cells. Statistical analysis was an Unpaired $t$-test with Welch's correction, $p$ values are two-tailed and shown above comparison. Data shown as individual biological replicates from three independent passage experiments ($n = 3$) ± SD. Source data for **e**, **j**, and **m** are provided in the Source Data file. The scale bar for **g** and **h** is 100 and 10 μm, respectively.

remained stable over three viral passages (Fig. 2m). Hence, the ZsGreen-expressing SARS-CoV-2 reporter virus generated by CPER was stable for at least three passages and could be used to facilitate fundamental studies requiring live virus detection in infected cells or animals and simplify screening of diagnostic tests, neutralizing antibodies, and antiviral drugs.

**Generation of RRV by CPER in mammalian and mosquito cells.** RRV is a mosquito-transmitted, positive-strand RNA virus that belongs to a globally distributed group of arthritogenic alphaviruses (family *Togaviridae*) that includes chikungunya (CHIKV), Mayaro, o'nyong nyong, and Sindbis viruses[33,34]. The acute disease manifestations associated with these viruses primarily involves polyarthralgia/polyarthritis, fever, rash, and/or myalgia, with arthritic manifestations often persisting for months[34]. RRV causes ~4000 cases of RRV disease in Australia annually, with an epidemic involving more than 60,000 cases in the Pacific Islands occurring between 1979 and 1980[33]. The 2004–2019 global CHIKV epidemic with greater than 10 million cases across more than 100 countries on four continents[33] further highlighted the ability of these alphaviruses to spread internationally[35]. The ability to easily manipulate alphavirus genomes should facilitate studies on inter alia virus–host interactions and evasion of antiviral response, thereby accelerating the development of new interventions.

Herein, the prototype RRV T48 isolate was used[36]. To establish a RRV CPER, primers were designed to generate six fragments covering the full genome of RRV T48 based on the published sequence (GenBank accession number GQ433359)[37] (Fig. 3a). Viral RNA was purified from the supernatant of RRV-infected *Aedes albopictus* mosquito (C6/36) cells and used to generate the first-strand cDNA, which was then used to amplify six fragments by PCR with 20–30 nucleotides overlaps (Fig. 3a, b). In addition, alpha-UTR-linker fragments were generated containing either the CMV or OpIE2 promoter, the first 22 nucleotides of the 5′UTR, the last 47 nucleotides of the 3′UTR, the polyA tract with 63 adenines and the HDVr (Fig. 3a and Supplementary Note 3).

Full-length cDNAs were assembled and circularized by CPER using six fragments covering the viral genome and the corresponding linker fragment, containing either the CMV promoter or the OpIE2 promoter. RRV viruses were recovered by transfecting the respective CPER output into HEK293T cells (CMV promoter) or C6/36 cells (OpIE2 promoter). RRV viruses recovered from HEK293T or C6/36 cells were passaged in C6/36 cells and were compared to the parental RRV T48 WT virus. HEK293T-derived and C6/36-derived CPER viruses produced similar mean plaques sizes in Vero cells to those produced by WT RRV viruses (Fig. 3c). In addition, similar replication kinetics of HEK293T-derived CPER virus and WT RRV virus were compared in Vero cells and were shown to be similar (Fig. 3d). Thus, CPER was successfully used to generate RRV in mammalian and mosquito cells directly from the viral RNA without any intermediate cloning.

**Utilizing CPER to investigate the role of E3/E2 furin cleavage site amino acids in RRV replication.** To demonstrate the utility of CPER methodology to modify the RRV genome, we generated two RRV mutants with changes in the furin-recognition site (NRSRHRR↓SV) located between E3 and E2 genes (Fig. 3e). Semliki forest virus (SFV) (GTRHRR↓TV) and H5N1 influenza virus hemagglutinin furin cleavage sites (ERRRKKR↓G) were incorporated instead of RRV cleavage site to generate RRV mutants RRV-SFV and RRV-H5, respectively (Fig. 3e). For each mutant, the 3′ and 5′ ends of the CPER fragments F4 and F5, respectively, were modified by PCR with primers incorporating the desired mutations (Fig. 3e). These modified fragments were then mixed with fragments F2, F3, and F6 and the alpha-UTR-linker fragment to generate a full-length circularized RRV cDNA by CPER. Mutant viruses were recovered from CPER-transfected HEK293T cells and were passaged in C6/36 cells. The presence of the desired mutations was confirmed by Sanger sequencing. RRV-H5 produced similar size plaques on Vero cells to the control CPER-recovered RRV (RRV-CPER WT), while the RRV-SFV formed smaller size plaques (Fig. 3f). RRV-H5 virus replicated

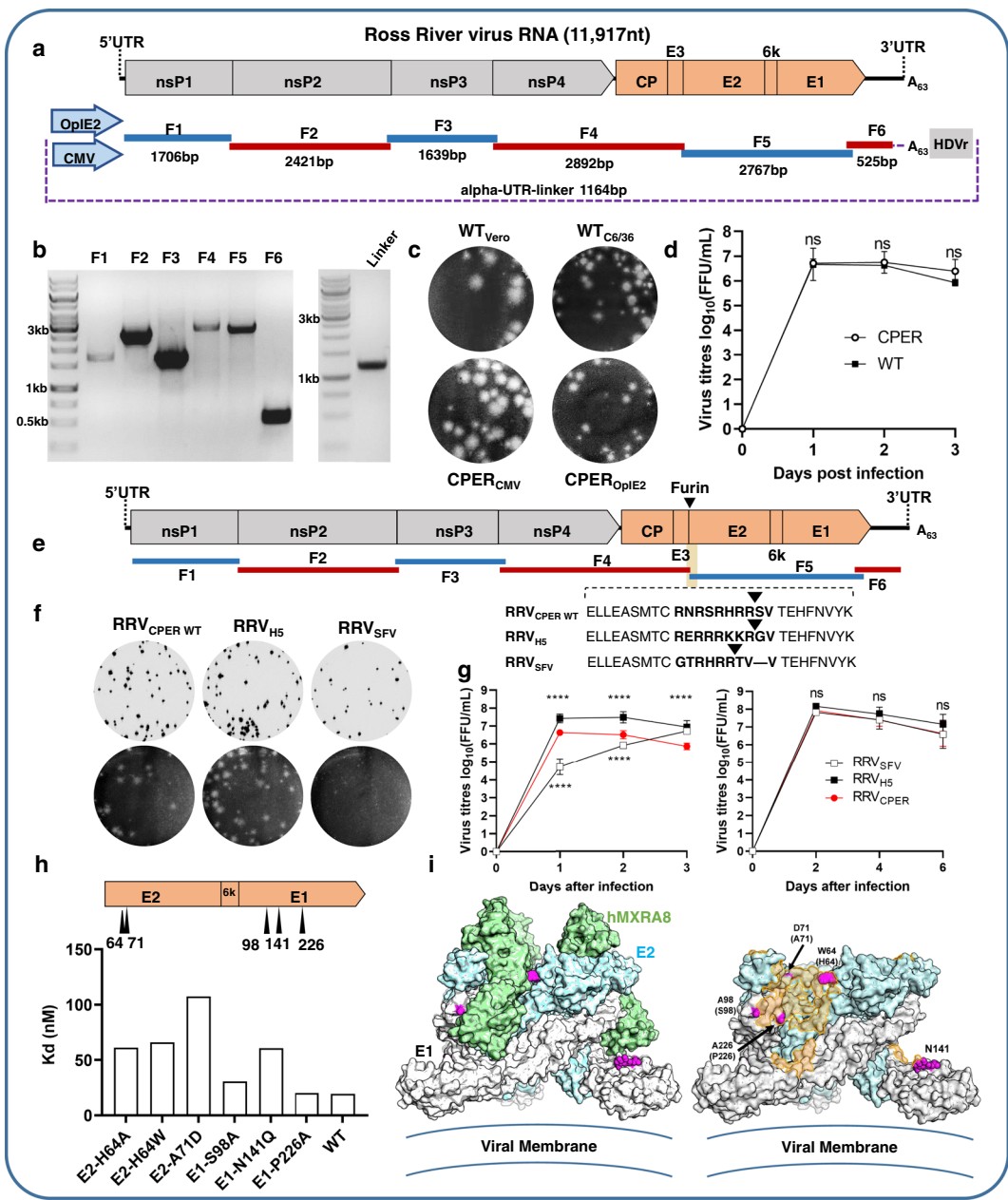

significantly more efficiently than RRV-CPER WT virus in Vero cells, while RRV-SFV virus replicated significantly less efficiently early (1 and 2 dpi) in infection (Fig. 3g). Both mutants replicated with similar kinetics to the RRV-CPER WT virus in C6/36 cells (Fig. 3g). The results show that changes in the furin cleavage site significantly affect the replication of RRV in mammalian cells but not in mosquito cells.

**Utilizing CPER to investigate the interactions of RRV with the alphavirus receptor MXRA8.** The cell adhesion molecule MXRA8 was recently identified as a receptor for multiple arthritogenic alphaviruses, including RRV and CHIKV[38]. Two subsequent papers explored the structural basis of the interaction between MXRA8 and CHIKV structural proteins. Cryo-electron microscopy and cocrystallography demonstrated that the MXRA8 ectodomain adopts an inverted, curved conformation that engages with both viral E1 and E2 proteins[39,40]. CHIKV E2 receptor binding determinants have been investigated using a variety of techniques[38–41]. Here, we used CPER to generate RRVs with site-specific mutations in E1 and E2 proteins to probe for receptor interactions (Fig. 3h, i). The presence of the desired mutations was confirmed by Sanger sequencing. Mutant viruses were grown at scale, gradient-purified, and purity confirmed by SDS-PAGE (Supplementary Fig. 5). A recombinant human MXRA8-Fc fusion protein was used to analyze whole virus–receptor interactions, with an increase in Kd (reduced binding affinity) observed for viruses with the E2 mutations H64A, H64W, and A71D (Fig. 3h). Thus, amino acids at these positions emerge to be essential for receptor interactions for both RRV and CHIKV[40]. However, the RRV-MXRA8 interaction did not tolerate W at position 64 (the residue present in CHIKV), indicating that the interactions between E2 residue 64 and MXRA8 residues D116/R196[40] may be distinct for the two viruses. Of the three E1 mutants, only N141Q showed a pronounced decrease in receptor binding, with this residue, an N-linked glycosylation site, conserved across the arthritogenic alphaviruses. The result suggests that glycosylation at this residue may play a role in modulating interactions with MXRA8. These experiments exemplify the utility of this CPER

**Fig. 3 Generation by CPER and characterization of wild-type and mutant Ross River viruses, and their application for studying virus–receptor interactions. a** Schematics of RRV T48 strain genome, overlapping fragments, and linker fragment used for CPER assembly. OpIE2—insect promoter, CMV—mammalian promoter. **b** Agarose gel electrophoresis of PCR-amplified RRV fragments and linker fragment. A representative of at least three experiments is shown. **c** Representative plaque morphologies of wild-type (WT) RRV viruses in Vero cells (top left) or *A. albopictus* C6/36 cells (top right) as well as of CPER-recovered RRV viruses generated using CMV promoter (bottom left) or OpIE2 promoter (bottom right). Infected cells were stained with crystal violet at 3 dpi. **d** Growth kinetics of WT (black square) and CPER-generated (white circle) RRV viruses in Vero 76 cells over a 3-day time course infected at MOI = 0.1, $n = 3$ independent experiments with two replicates in each, statistical analysis to compare each time point for each of the mutants against CPER WT virus was performed by two-way analysis of variance with Tukey's multiple comparisons test. Adjustments were not made for multiple comparisons. Mean values of the three independent experiments ± SD of the mean are shown. **e** Schematics of overlapping RRV fragments amplified from RRV$_{T48}$ cDNA and introduced furin site mutations at the E3/E2 cleavage junction. Modified furin cleavage sites are introduced from Influenza virus H5N1 (RRV$_{H5}$) and Semliki forest virus (RRV$_{SFV}$) through the overlapping regions in fragments 4 and 5. Location of furin cleavage is indicated by an arrowhead. **f** Representative foci and plaque morphologies of CPER-generated RRV furin site mutants as shown from immuno-plaque assay at 12 hpi (top) and crystal violet staining at 3 dpi (bottom). **g** Growth kinetics of CPER-generated RRV furin mutants (RRV$_{SFV}$; white square, RRV$_{H5}$; black square, RRV$_{CPER}$; red circle), in mammalian (Vero) cells (left) and *A. albopictus* C6/36 cells (right), $n = 3$ independent experiments with two replicates in each. Statistical analysis was performed by two-way analysis of variance with Tukey's multiple comparisons test. Adjustments were not made for multiple comparisons. ****$p \leq 0.0001$. Mean values of the three independent experiments ± SD of the mean are shown. **h** Binding assay for E1/E2 RRV mutants using purified virus and recombinant hMXRA8 receptor and represented as equilibrium-binding affinity (Kd) at 37 °C. Data used to generate the nonlinear curve are available from Source Data file, data represent two independent experimental replicates ($n = 2$). Summary table presented here as bar graph. **i** Position of the RRV mutants shown in **h** on the reported CHIKV-hMXRA8 structure (PDB:6J08). The heterotrimers of E1 and E2 are shown in white and blue surface representation, respectively. hMXRA8 is shown in green surface representation in the left panel, and the binding footprint shown in orange on the right. Mutated residues are indicated in pink, with labels indicating the amino acid for CHIKV at each position and corresponding residue for RRV in brackets. Source data for **d**, **g**, and **h** are provided in the Source Data file.

approach to investigate the role of individual residues in key host–virus interactions using replication-competent viruses.

**Generation of murine and HuNoVs by CPER.** HuNoVs are positive-strand RNA viruses belonging to the Norovirus genus in the *Caliciviridae* family. They are a major cause of acute gastroenteritis in developing and developed countries, causing ~220,000 deaths annually[42]. Research endeavors seeking to develop interventions to control HuNoV outbreaks are severely hampered by the inability to cultivate HuNoVs in the laboratory. Recent studies have shown that HuNoV can replicate (i) in B-cell-like cell lines only when cocultured with specific enteric bacteria[43], (ii) in enteric organoids[44], or (iii) in zebrafish larvae[45]. The closely related genogroup V MNV remains the most widely used virus for norovirus research[46]. Noroviruses have a ~7.5 kb genome that encodes for nine or ten proteins, which have various roles in replication of the viral genome, polyprotein cleavage, translation, host manipulation, and assembly of virus particles[47]. The RNA genome is covalently attached to the viral protein VPg at its 5′ end and is polyadenylated at the 3′end[47].

CPER for MNV was generated from fragments amplified from the plasmid DNA pSPORT-T7-MNV1 containing full-length MNV cDNA clone under control of T7 promoter[48]. Three fragments covering the entire MNV genome and containing 27–34 nucleotide overlaps were PCR-amplified, assembled into full-length cDNA, and circularized with a linker fragment containing polyA tail with 30 adenines (Fig. 4a, b and Supplementary Note 2) by CPER. As NIH3T3 cells are more efficiently transfected with DNA than RAW264.7 cells, CPER was first transfected into NIH3T3 cells (P$_0$), generating ~2.5 × 10$^2$ PFU/mL by 3 dpt, and then amplified in RAW264.7 cells (P$_1$) generating ~10$^8$ PFU/mL by 3 dpi (Fig. 4c). CPER-generated virus formed similar size plaques to the original MNV isolate in RAW264.7 cells (Fig. 4d). Western blot of CPER-generated virus with anti-MNV capsid (VP1) antibodies detected the expected 57 kDa band representing full-length MNV capsid protein (Fig. 4e).

To assess whether CPER could be used to generate HuNoV from clinical samples, total RNA was purified from a deidentified patient fecal sample, and cDNA was synthesized using an oligo (dT) primer. This cDNA was used to amplify three fragments covering the HuNoV genome (Fig. 4a, b). CPER was conducted

using the same protocol and cycling conditions for MNV, followed by transfection into NIH3T3 cells. Viral RNA was then purified from cell supernatants at 3 and 72 hours post transfection (hpt), reverse transcribed and quantified by real-time PCR. A clear increase in viral RNA was observed between 3 and 72 hpt (Fig. 4f), indicating virus assembly and release. No viral RNA was detected in the mock samples (Fig. 4f). Overall, the results show the successful generation of murine and HuNoVs by CPER and illustrate the utility of CPER as a reverse genetics system for generating replication-competent virus from clinical samples.

**Generation of Casuarina virus by CPER.** Casuarina virus (CASV, Alphamesonivirus 4) is an insect-specific virus belonging to the *Mesoniviridae* family, order *Nidovirales*[49]. Mesoniviruses have a positive-sense single-stranded 20–21 kb RNA genome, with a comparable gene organization to the coronaviruses[50], which also belong to the order *Nidovirales*. The virion architecture of mesoniviruses is similar to that of the coronaviruses, with large globular heads attached to low-density stalks protruding from the enveloped virion[49,51]. In contrast, mesoniviruses replication is restricted to mosquitoes, with no ability to replicate in vertebrate cells[49,52]. The factors determining mosquito host specificity of CASV remain unknown. A simple and easy-to-use CASV reverse genetic system is needed to gain insights into the role of these factors.

As for SARS-CoV-2, the large size of the CASV genome (~20 kb) complicates conventional reverse genetics methods. We, therefore, developed the CPER methodology as a simple reverse genetics system for manipulating the CASV genome. Five fragments encompassing the complete CASV genome with 30–43 nucleotides overlaps were amplified from viral cDNA (Fig. 4h). A linker fragment was also generated (Fig. 4g, h and Supplementary Note 4) that contained the OpIE2 promoter, the first 25 nucleotides of the 5′UTR, the last 40 nucleotides of the 3′ UTR, a polyA tail with 53 adenines, and the HDVr. The full-length CASV cDNA was then assembled and circularized by CPER. Three different CPER conditions that differed in the length of extension time and the number of cycles were tested, with all of them yielding infectious virus. The relative efficiencies of different CPER conditions and the identity of CASV in

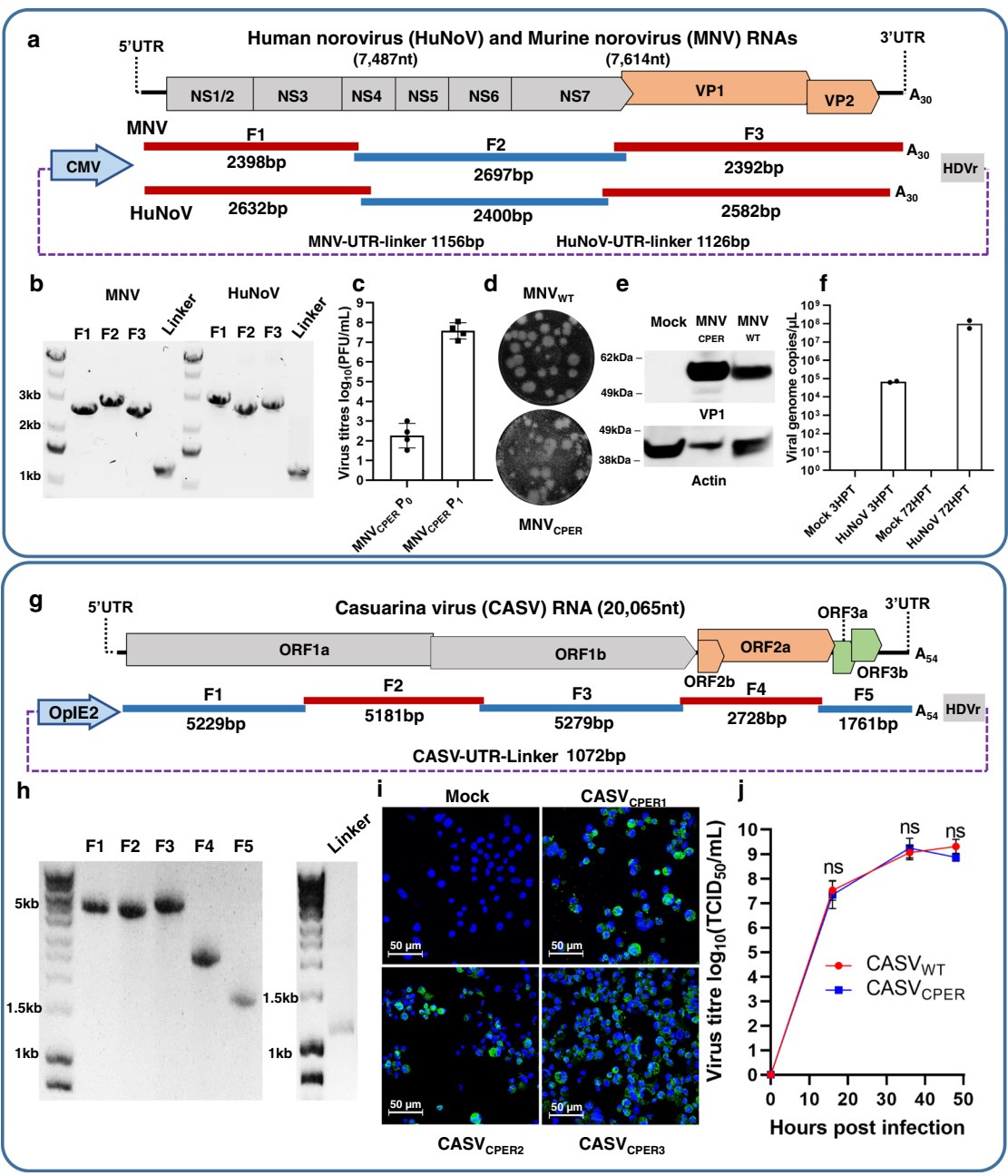

**Fig. 4 Generation of mouse and human noroviruses and insect-specific mesonivirus CASV by CPER and their characterization. a** Schematics of human norovirus (HuNoV) and murine norovirus (MNV) genomes and of overlapping fragments and linker fragment used for CPER assembly. **b** Agarose gel electrophoresis of PCR-amplified MNV, HuNoV, and linker fragments showing a representative image of two experiments for HuNoV and at least three for MNV. **c** Virus titers quantitation of MNV CPER-transfected NIH3T3 cells at 3 days post transfection (MNV$_{CPER}$ P$_0$) and of virus further amplified in RAW264.7 cells at 3 days post infection (MNV$_{CPER}$ P$_1$), results indicate four independent transfections and subsequent infections. **d** Representative plaque morphologies of wild-type MNV (MNV$_{WT}$) and CPER-recovered MNV (MNV$_{CPER}$). **e** Immunoblot analysis of mock, MNV$_{WT}$ and MNV$_{CPER}$ infected NIH3T3 cell lysates probed with anti-MNV-VP1 (top) and anti-actin (bottom) antibodies blot are representative of two experiments. **f** HuNoV genomic RNA (gRNA) quantitation by RT-qPCR analysis of supernatants from CPER-transfected NIH3T3 cells at 3 and 72 h post transfection from two independent experiments (n = 2). **g** Schematics of Casuarina virus (CASV) genome and overlapping fragments as well as linker fragment used for CPER assembly. **h** Agarose gel electrophoresis of PCR-amplified CASV fragments and linker fragments showing a representative image of more than three independent experiments. **i** Immunofluorescence analysis by confocal microscopy of *A. albopictus* C6/36 cells transfected with CPER products generated by three different CPER cycling conditions (CPER1, CPER2, CPER3), showing representative images of one independent experiment. CPER3 has been reproduced in two separate independent experiments. The scale bar is 50 μm. Viral antigen is visualized by staining with anti-CASV ORF2a protein monoclonal antibody C.9D7 and cell nuclei stained with Hoechst 33342. **j** Growth kinetics of CPER-generated (blue square) and WT CASV (red circle) in C6/36 cells infected at MOI = 0.1, n = 2 independent experiments were conducted with three replicates. Statistical analysis was performed by two-way analysis of variance with Tukey's multiple comparisons test. Graphs in **c**, **f**, and **j** are mean values ± SD. Source data for **c**, **f**, and **j** are provided in the Source Data file.

transfected C6/36 cells were shown by immunofluorescence assay (IFA) using an anti-CASV monoclonal antibody (Fig. 4i). CPER-recovered CASV replicated with comparable efficiency to the WT CASV in C6/36 cells (Fig. 4j). The results demonstrate the utility of CPER as a robust reverse genetic system for CASV.

## Discussion

Herein, we established a CPER-based methodology for the construction of SARS-CoV-2 and CASV and demonstrated the utility of this method as a versatile reverse genetics platform for polyA tail-containing viruses with large genomes. Six or seven PCR-generated fragments, up to 5.5 kb in length with ~20–40 nucleotides overlap, are assembled in vitro into a full-length circular DNA with the aid of a linker fragment that contains eukaryotic expression promoter upstream of the viral 5′end and polyA tail downstream of the viral 3′end followed by the HDV ribozyme. The incorporation of a polyA tail at the end of the viral 3′UTR and immediately prior to the HDV ribozyme allows to generate authentic viral 3′end with polyA tail, while SV40 polyA signal downstream of HDVr ensures efficient transcription termination[8,53]. The 30 and 20 kb viral RNA genomes are then generated in the nucleus of cells transfected with CPER-generated DNA and exported into the cytoplasm, where they initiate the generation of authentic viruses, i.e., viruses with replicative and pathogenic characteristics (for SARS-CoV-2) indistinguishable from WT viruses. Remarkably, the CPER method described herein reliably and rapidly produces authentic viruses despite (i) the large genome sizes that would likely exacerbate misassembly errors, (ii) the presence of nuclear RNA splicing activities[54], (iii) the multiple processes required for nuclear export and translation of RNA[55], and (iv) the presence of host cell enzymes that degrade viral RNA[56,57]. Thus, genome size does not appear to be a limitation for the CPER methodology, which bodes well for future studies of RNA viruses with larger genomes, such as the recently discovered planarian secretory cell nidovirus that has a ~41 kb genome[58]. The ability to recover by CPER RNA viruses that have large genome and that contain polyA tail represent a significant advance from our previous CPER developments for flaviviruses, which have relatively small (11 kb) genome and do not contain polyA tail.

An alternative reverse genetic system has been developed and used for generating pathogenic coronaviruses and for manipulating the SARS-CoV-2 genome, and is in some ways similar to CPER[15,17,24,26]. The method involves an in vitro ligation for assembling full-length cDNA from a set of cloned cDNA fragments. RNA is then generated by in vitro transcription and is then transfected into cells to generate an infectious virus. The method involves cloning each viral cDNA fragment into a plasmid vector and propagation in bacteria, whereas CPER generates these fragments by RT-PCR. Some of the SARS-CoV-2 fragments in the in vitro ligation protocol were unstable when cloned and propagated in bacteria and therefore required the use of low copy number plasmid vectors, different *Escherichia coli* strains, and lower incubation temperature for more stable propagation[24]. Similar instability for some of the SARS-CoV-2 fragments was also encountered in YAC reverse genetics system[16]. The fragments for the in vitro ligation method are then liberated from the amplified plasmids using class IIS restriction enzymes, with unique recognition sites engineered at the end of each fragment. Fragments are then ligated using T4 DNA ligase by cohesive ends left after restriction enzyme digestion. In contrast, the CPER method anneals fragments by ~20–40 nucleotide overlaps, uses DNA polymerase to fill in the remaining fragment sequences, and generates circular dsDNA ready for transfection, all in one reaction. After transfection of the completed CPER reaction, the

method utilizes cellular enzymes to repair the remaining "nicks" in the dsDNA and to transcribe the viral RNA[6–8]. Therefore, the primary advantages of the CPER approach are that the steps of engineering special restriction sites, cloning, bacterial propagation, in vitro ligation, and in vitro RNA transcription are not required.

Generation of viral mutants by the CPER methodology described herein is a simple and rapid process and is exemplified for SARS-CoV-2 (Fig. 2a–e) and RRV (Fig. 3e–i). For example, to generate the D614G mutant of SARS-CoV-2, fragment F5 was replaced with two subfragments F5A and F5B. F5A was generated using the same forward primer as for F5 and a reverse primer containing the nucleotide substitution coding for the new amino acid. F5B was generated by using the same reverse primer as for F5 and a forward primer that is complementary to the aforementioned primer with the substitution. The subfragments were then used with the rest of the fragments in the CPER assembly. Generating recombinant viruses with deletions and/or insertions can be achieved by generating subfragments with appropriate primers. For example, to develop the SARS-CoV-2-ΔORF7a-ZsGreen virus, the ZsGreen reporter gene was generated by replacing fragment F6 with two subfragments, F6A and F6B, flanking the ORF7a deletion, and adding the ZsGreen fragment containing overlapping ends with F6A and F6B subfragments (Fig. 2f). Conceivably, multiple mutations, deletions and/or insertions can be simultaneously introduced by CPER by swapping multiple WT fragments with corresponding fragments containing the desired mutations/deletions and/or by adding desired insertion fragments. However, we have not explored the upper limit of the number of cDNA fragments that can be successfully assembled using CPER, with each mutation/insertion requiring the addition of another fragment to the CPER reaction. If multiple mutations are needed in one fragment, this fragment could be generated synthetically and incorporated into the CPER reaction. We have shown previously that an infectious virus can be generated entirely from synthetic DNA fragments using CPER[7]. If mutations are needed in multiple fragments, a mixture of synthetic and split fragments (e.g., F5a and F5b in Fig. 2a) might be envisaged. Clearly, cDNA fragments derived from one virus isolate can also be used together with cDNA fragments from another virus isolate (and/or synthetic fragments) to generate viruses with a combination of desired mutations[8]. Synthetic fragments could also, for instance, be used to incorporate clusters of repeated sequences (e.g., microRNA target sites) as long as primers to amplify the fragments are located outside the repeat regions.

The versatility of the CPER methodology was further illustrated herein by generating five polyA tail-containing positive-strand RNA viruses from four different virus families. In addition, the generation of viruses using insect cells instead of mammalian cells was illustrated for RRV (Fig. 3) and CASV (Fig. 4g–j) by incorporating an insect-specific promoter instead of a mammalian expression promoter for transcription of viral RNA. CPER was also applied to generate virus directly from RNA purified from a clinical sample, a process illustrated herein for HuNoV (Fig. 4a). This is particularly useful for viruses like HuNoV, which cannot be readily isolated by in vitro culture.

While the CPER method is rapid and robust, it does have certain limitations. It relies on efficient DNA transfection; hence, the highly transfectable HEK293T cells provide an optimal choice. However, HEK293T cells do not support robust SARS-CoV-2 infection. Hence, the coculture of CPER-transfected HEK293T cells with Vero E6 was used to recover high virus titers. Propagation of SARS-CoV-2 in Vero E6 cells can lead to the selection of mutations[19,22], an issue seen in other reverse genetics systems[16,21,27] and observed herein for CPER-generated

viruses. Our data support the view that choice of cell lines for SARS-CoV-2 recovery/amplification such as ACE2- and/or TMPRSS2-expressing cells[21,23] may reduce the selection of unintended mutations. In addition, using a lower passage virus with more homogeneous sequence as the source of cDNA to amplify fragments for CPER assembly could further alleviate the selection of minor variants. On the other hand, the ability of CPER to faithfully recapitulate viral quasi-species[6] may be beneficial for studies on virus adaptation and evolution.

In summary, we have established a simple, rapid, and versatile CPER-based reverse genetic platform for a range of positive-strand RNA viruses with polyA tails. We have demonstrated its practical utility in various settings, particularly its application for viruses with large genomes like SARS-CoV-2. As new SARS-CoV-2 variants continue to emerge[32,59,60], the ability to easily and rapidly manipulate such a large viral genome by CPER will greatly expedite identifying new determinants of transmission and pathogenesis.

## Methods

**Cells**. Human embryonic kidney 293T cells (HEK293T), African green monkey kidney cells Vero E6 and Vero 76 cells, murine macrophage RAW264.7, and murine fibroblast NIH3T3 cells were maintained at 37 °C in Dulbecco modified Eagle medium (DMEM; Gibco) supplemented with 10% fetal calf serum (FCS; Gibco), 1% GlutaMAX (Gibco), and 1% sodium pyruvate (100 mM; Gibco). *A. albopictus* C6/36 cells were grown at 28 °C in Royal Park Memorial Institute (RPMI) medium (Gibco, USA), 10% FCS (Gibco), and 1% GlutaMAX (200 mM; Gibco). Following viral infections, cells were maintained in corresponding media with 2% FCS and 10,000 U/mL of penicillin and 10,000 μg/mL of streptomycin (Gibco, USA).

Vero E6-TMPRSS2 cells were generated by transduction with lentivirus containing puromycin-selectable codon-optimized human TMPRSS2 construct and validated with anti-TMPRSS2 antibody (Abcam, ab109131) (full details are available in Supplementary Note 1). HEK293T-hACE2 cells were provided by Jesse Bloom (Fred Hutchinson Cancer Research Centre, Washington, USA)[18].

**Viruses**. The SARS-CoV-2 isolates were sequenced by Dr D. Warrilow (Queensland Health, Brisbane, Australia), hCoV-19/Australia/QLD02/2020 (QLD02) (GISAID Accession ID; EPI_ISL_407896) and hCoV-19/Australia/QLDID935/2020 (QLD935) (GISAID Accession ID: EPI_ISL_436097), named here as QLD02 and QLD935, respectively. The viruses were isolated from patient nasopharyngeal aspirates via inoculation of Vero E6 cells and virus stocks were then produced in Vero E6 cells. This work received ethical clearance from the Queensland Health Forensic and Scientific Services Human Ethics Committee (EC00305), approval reference HEC 21-08. Passage 3 or 4 of QLD02 isolate on Vero E6 cells was used for purifying viral RNA and generating cDNA. All the work with infectious SARS-CoV-2 and CPER-generated SARS-CoV-2 recombinants not involving animals was performed in a certified PC3 facility at The University of Queensland (UQ) and approved by the UQ Institutional Biosafety Committee (UQ IBC, approvals IBC/390B/SCMB2020 and IBC/1301/SCMB/2020). Work with RRV, CASV, and corresponding CPER-recovered viruses was performed in PC2 facilities at SCMB, UQ, and approved by UQ IBC (IBC/1205/SCMB/2018 and IBC/1289/SCMB/2020, respectively). Generation of MNV/HuNoV CPER was approved by the University of Melbourne Institutional Biosafety Committee (approval IBC 2019.035). Ross River virus prototype strain T48 was initially isolated from *Aedes vigilax* mosquitoes collected near Ross River (Townsville, Australia) in 1959[61] and a stock produced in C6/36 cells. MNV1 was obtained from H. Virgin (Washington University School of Medicine) and propagated in RAW264.7 cells. Casuarina virus strain 0071 (GenBank accession number: NC_023986) was originally isolated from *Coquillettidia xanthogaster* mosquitoes collected in Darwin in 2010[49], and a virus was passaged 1–2 times in C6/36 cells to produce viral stock for RNA extraction.

**Viral RNA extraction**. For SARS-CoV-2 and RRV, 15 mL of supernatant from infected cells were concentrated to ~250 μL by using the Amicon 100 kDa column (Merck Millipore, USA). RNA from the concentrated virus was extracted using TRIzol LS reagent (Thermo Fisher Scientific, USA) as per the manufacturer's protocol. For HuNoV, a deidentified HuNoV GII.4 positive fecal specimen (10% w/v in Hanks media) was provided by the Victorian Infectious Diseases Reference Laboratory under human ethics approval HESC 1749300.1. Viral RNA was extracted using TRIzol reagent (Invitrogen, USA) as per the manufacturer's protocol. For CASV, viral RNA was extracted from the culture supernatant of infected C6/36 cells using a NucleoSpin RNA virus kit (Macherey-Nagel).

**cDNA synthesis**. For SARS-CoV-2, viral RNA was used to prepare cDNA with Protoscript II first-strand cDNA synthesis kit and random primer mix containing a mixture of hexamers and an anchored-dT primer ($dT_{23}VN$) as per manufacturer's protocol (New England Biolabs, USA). All primers used in this study are shown in Supplementary Table 3 with individual primer names referred to throughout the text. For RRV, viral RNA was used to prepare cDNA with an RRV-3′UTR- reverse primer and SuperScript III First-Strand Synthesis System (Thermo Fisher Scientific, USA). For HuNoV, viral RNA was used to prepare cDNA with Oligo(dT)23 primer (Sigma-Aldrich) and First-Strand Synthesis System (Thermo Fisher Scientific, USA). For CASV, viral RNA was used as a template for cDNA synthesis using SuperScript IV (Thermo Fisher Scientific) with priming by random hexamers (Promega, USA).

**PCR amplification of DNA fragments for CPER assembly**

*SARS-CoV-2 fragments*. Six SARS-CoV-2 fragments (Fig. 1a) were amplified from viral cDNA using high-fidelity Prime Star GXL DNA polymerase (Takara Bio) and corresponding pairs of primers. For generating D614G virus, two subfragments of fragment 5, 5A, and 5B (Fig. 2a) were generated from viral cDNA with primers incorporating GAT to GGT change at codon 614 in the spike gene. For generating ZsGreen reporter virus, two subfragments of fragment F6 were generated from viral cDNA, subfragment F6A ending at codon 13 in the open reading frame 7 (ORF7a) and subfragment F6B starting at codon 109 of ORF7a (Fig. 2f). ZsGreen with the addition of overlapping regions for subfragments F6A and F6B (Fig. 2f) was generated by PCR from pCCI-SP6-ZIKV-ZsGreen, kindly provided by Dr A. Merits (Tartu University, Tartu, Estonia).

*RRV fragments*. Six overlapping fragments (Fig. 3a) were amplified from RRV viral cDNA using Q5 High-Fidelity DNA Polymerase (New England Biolabs, USA). To generate the furin site mutants (Fig. 3e), the 3′ and 5′ ends of the CPER fragments F4 and F5 were modified in two steps. Step 1, new fragments F4 and F5 that flank the furin cleavage sites were amplified from cDNA. Step 2, fragments F4 and F5 from the first step were used as a template to create new amplicons. To create amino acid mutations in E2 and E1 genes (Fig. 3h), the new CPER fragments F4 and F5 for E2 mutations and F5 and F6 for E1 mutations were generated from the viral cDNA by Q5 High-Fidelity DNA polymerase.

*HuNoV and MNV fragments*. Three overlapping HuNoV fragments (Fig. 4a) were amplified from viral cDNA, while the three overlapping MNV fragments (Fig. 4a) were amplified from the full-length MNV clone pSPORT-T7-MNV1[48]. High-fidelity Prime Star GXL DNA Polymerase (Takara Bio) was used for all PCR amplifications.

*CASV fragments*. Five overlapping fragments (Fig. 4g, h) were amplified from viral cDNA using high-fidelity Prime Star GXL DNA polymerase (Takara Bio).

**Linker fragments**. Linker fragments for each viral CPER assembly were generated as described in Supplementary Notes.

**CPER reactions, transfections, and virus recovery**

*SARS-CoV-2*. Purified SARS-CoV-2 cDNA fragments and SARS-CoV-2 linker fragment (Supplementary Note 2) were mixed in equimolar amounts (0.1 pM each) in a 50 μL reaction volume containing 200 μM of each dNTP, 1x PS GXL reaction buffer, and 2 μL of Prime Star GXL DNA polymerase. The following cycling conditions were used: initial denaturation at 98 °C for 30 s, followed by 12 cycles of denaturation at 98 °C for 10 s, annealing at 55 °C for 20 s, and extension at 68 °C for 10 min, followed by a final extension at 68 °C for 10 min. For the generation of the D614G mutant, subfragments F5A and F5B were used together with other SARS-CoV-2 fragments and the linker fragment in CPER reaction using the same cycling conditions. For generating ZsGreen reporter virus, subfragments F6A and F6B and ZsGreen fragment were used together with the other SARS-CoV-2 fragments and the linker fragment in the CPER reaction using the same cycling conditions. CPER reactions were then used for transfections without any purification. Two independent transfection experiments (CPER1 and CPER2) were performed by transfecting HEK293T cells with 50 or 25 μL of CPER reactions in a 6-well or a 12-well plate, respectively, using Lipofectamine LTX Plus reagent (Invitrogen, USA) as per manufacturer's protocol. Six hours post transfection, cells were trypsinized and transferred to a confluent monolayer of Vero E6 cells in a six-well plate. Supernatants containing viruses were harvested when cytopathic effect (CPE) was pronounced (6–9 days post transfection, reaching titers of up to ~$10^6$ FFU/mL as determined by iPA described below) and amplified once on Vero E6 cells to generate viral stocks with titers of ~$5 \times 10^6 - 10^7$ FFU/mL. CPER3 virus was generated by transfecting HEK293T-ACE2 cells in a six-well plate with CPER mix using Lipofectamine LTX Plus reagent. This resulted in the recovery of $1.8 \times 10^5$ FFU/mL CPER3 virus in the culture fluid by day 8 after transfection (P0). The secreted virus was then amplified in T75 flask of Vero E6 cells for 3 days and reached the titer of $2.3 \times 10^7$ FFU/mL (P1).

For recovery of D614G mutant virus and dORF7a-ZsGreen (ZsGreen) reporter viruses, CPER fragments were amplified from passage 4 QLD02 cDNA (D614G) or passage 3 QLD02 cDNA (ZsGreen) and assembled by CPER as above. CPER reactions (50 μL) were then transfected into HEK293T cells in a six-well plate using Lipofectamine LTX Plus reagent, and at 6 hpt cells were trypsinized and transferred

to a T75 flask of confluent Vero E6 cells (for D614G and ZsGreen1) or Vero E6-TMPRSS2 cells (for ZsGreen2). Viruses were harvested from supernatants when CPE was pronounced (6–8 days post transfection), and virus titers were $1.1 \times 10^7$ FFU/mL for D614G mutant, $2.5 \times 10^6$ FFU/mL for ZsGreen1 virus, and $5 \times 10^6$ FFU/mL for ZsGreen2 virus. No additional amplification in Vero E6 cells was performed for these viruses.

*RRV*. Equimolar amounts (0.1 pM each) of six RRV RT-PCR fragments and either CMV promoter- or OpIE2 promoter-containing alpha-UTR-linker fragments (Supplementary Note 3) were assembled into a circular full-length cDNA using the Q5 High-Fidelity DNA Polymerase (New England Bio Lab, USA) with the following conditions: 98 °C for 2 min followed by 12 cycles of 98 °C for 10 s, 60 °C for 15 s, 68 °C for 10–13 min, and a final extension at 68 °C for 10–13 min. Subsequently, the CPER reactions were transfected directly without any purification into HEK293T cells (CMV promoter) or C6/36 cells (OpIE2 promoter). Transfection into HEK293T cells was performed using the Lipofectamine LTX Plus reagent (Invitrogen, USA), while transfection into C6/36 cells was performed using TransIT-LT1 reagent (Mirus Bio, USA). Viruses were harvested from the supernatants of transfected HEK293T or C6/36 cells and amplified on C6/36 cells to generate viral stocks. Virus titers were determined by iPA with mouse anti-E1 monoclonal antibody G8[62].

*HuNoV and MNV*. For both HuNoV and MNV, 0.1 pM of each of the three respective cDNA fragments and corresponding linker fragments (Supplementary Note 2) were combined in CPER reaction with Prime STAR GXL DNA polymerase (Takara Bio) and subjected to the following cycling conditions: 98 °C for 30 s, followed by 12 cycles of 98 °C for 10 s, 55 °C for 30 s, 68 °C for 10 min, and a final extension at 68 °C for 10 min. The entire CPER reaction was transfected with Lipofectamine 3000 into NIH3T3 cells seeded in a well of a six-well plate. At 3 and 72 hpt, cell supernatant was collected, clarified from cell debris by centrifugation at $500 \times g$ for 5 min and stored at −80 °C for passaging, plaque assays and RT-qPCR.

*CASV*. Equimolar amounts (0.1 pmol each) of the five CASV fragments and the CASV OpIE2 linker fragment (Supplementary Note 4) were used in a CPER with Prime Star GXL DNA polymerase (Takara Bio) in a 50 μL reaction. Three different cycling conditions were used to produce CPER1, CPER2, and CPER3 reactions. The CPER1 reaction was cycled at 98 °C for 2 min, followed by 20 cycles of 98 °C for 10 s, 55 °C for 15 s, 68 °C for 25 min, and a final extension of 68 °C for 25 min. The CPER2 reaction was cycled at 98 °C for 2 min, followed by 35 cycles of 98 °C for 10 s and 68 °C for 15 min, and a final extension at 68 °C for 25 min. The CPER3 reaction was cycled at 98 °C for 2 min, followed by 35 cycles of 98 °C for 10 s, 55 °C for 15 s, and 68 °C for 15 min, followed by a final extension at 68 °C for 15 min. Subsequently, 25 μL of the 50 μL CPER reaction was transfected into C6/36 cells in a six-well plate using TransIT-LT1 (Mirus Bio), according to the manufacturer's instructions. At 7 days post transfection, the supernatant was harvested and passaged onto C6/36 cells. Immediately following the removal of the supernatant, transfected cells were assessed by IFA.

**Nanopore sequencing and bioinformatic analysis**. For whole-genome sequencing of SARS-CoV-2 isolates, CPER-generated viruses, and zsGreen reporter viruses, the nCoV-2019 sequencing protocol v3 (Josh Quick, University of Birmingham) was used with minor modifications. Briefly, cDNA generated from RNA isolated from cell culture supernatant was amplified using with ARTIC network v2 primers using two-step PCR amplification with Q5® High-Fidelity DNA Polymerase (New England Biolabs, USA). For amplification of whole-genome zsGreen sequencing, the ORF7_zsGreen primer pair was spiked into PCR reactions (0.030 μM) to recover bridging amplicons up and downstream of the ORF7a replacement. For sequencing of CPER DNA, the CPER fragments were used directly without additional PCR amplification.

PCR fragments were purified using AMPure XP beads (Beckman Coulter, USA) and subjected to End Repair/dA-Tailing using the NEBNext® Ultra™ II Module (New England Biolabs, USA) and multiplexed using the Native Barcoding Expansion kit (EXP-NBD104, Oxford Nanopore, UK) and Ligation Sequencing Kit (SQK-LSK109, Oxford Nanopore, UK). Prepared libraries were then quantified using the Qubit™ dsDNA HS Assay Kit (Thermo Fisher Scientific, USA) and loaded into equimolar concentrations totaling 20 fmol into a Flongle flow cell (FLO-FLGOP1, Oxford Nanopore, UK). Base-calling and adapter trimming of fast5 files were performed using the guppy_basecaller (Version: 3.3.0+ef22818). Clean fastq base-called files were subsequently mapped to the SARS-CoV-2 isolate QLD02 (GISAID accession EPI_ISL_407896) using the Burrows–Wheeler Aligner (Version: 0.7.13-r1126)[63] using the flags (mem -x ont2d). Depth of coverage of mapped alignment files was determined using samtools (v1.3) depth. Single-nucleotide variants of alignment files were identified using iVar (v1.2.2)[64] with a minimum quality score threshold of 20. Coverage and frequencies of variant positions were visualized and calculated using Integrative Genomics Viewer (Version: 2.7.0)[65]. Base-called fastq data generated in this study have been deposited in the Sequence Read Archive hosted by the National Center for Biotechnology Information with accession number PRJNA707404.

**Standard plaque assays**. SARS-CoV-2 and RRV plaque assays were conducted on Vero E6 and Vero 76 cells, respectively. The overlay medium for both contained 0.375% low-melting point agarose (Bio-Rad, USA). Cells were fixed at 2 or 3 dpi with 4% formaldehyde and stained with 0.2% crystal violet solution. RAW264.7 cells were seeded onto 12-well plates and incubated at 37 °C for 24 h until 70% confluent. Cell culture supernatants containing virus were tenfold serially diluted in DMEM, inoculated onto cell monolayer in duplicate, and incubated at 37 °C for 1 h. Following inoculation, overlay medium (70% DMEM, 2.5% [vol/vol] FCS, 13.3 mM NaHCO₃, 22.4 mM HEPES, 200 mM GlutaMax, and 0.35% [wt/vol] low-melting-point agarose) was added to each well, solidified at 4 °C, and incubated at 37 °C. After 48 h, cells were fixed in 10% formalin for >1 h and overlay removed. Fixed-monolayers were stained with 0.2% crystal violet (80% PBS, 20% methanol) for 30 min and plaques enumerated[66].

**Immuno-plaque assays**. Viral titers were determined using an optimized iPA for SARS-CoV-2[67]. Briefly, $4 \times 10^4$ cells per well of Vero E6 were seeded in 96-well plates and incubated for 14 h to reach 100% confluence. Samples were tenfold serially diluted in DMEM supplemented with 2% FCS and 10,000 U/mL of penicillin and 10,000 μg/mL of streptomycin (Gibco, USA) and 25 μL of each dilution incubated with the cells for 30 min at 37 °C. Then, 175 μL of overlay medium was added to each well without removing the inoculum. After 14 h post infection, the overlay medium was removed, and the monolayer was fixed by adding cold 80% acetone and kept for 30 min at −20 °C. Next, acetone was removed, and the plate was fully dried for 2 h. Subsequently, viral foci in fixed infected cells were detected using the mouse monoclonal antibody CR3022[67]. The plates were first blocked for 60 min at 37 °C by adding 150 μL of blocking solution Pierce Clear Milk Blocking buffer (Thermo Fisher Scientific Inc, USA, Cat. No. 37587) to 96-well plates. Following, fixed cells were probed with primary mAb for 1 h at 37 °C, using 50 μL/well (50 μg/well). After that, plates were washed five times with phosphate buffered saline (PBS) containing 0.05% Tween-20 with 5 min incubating in between each wash. Subsequently, a fluorophore-conjugated goat anti-mouse IgG secondary antibody (IRDye® 800CW, LI-COR, USA, Cat. No. 926-32210) was added to each well (20 ng/well) and incubated for 1 h at 37 °C. Then, plates were washed five times and dried before scanning. Finally, plates were scanned using the LI-COR Biosciences Odyssey Infrared Imaging System (Odyssey CLx, Li-COR, USA). Image acquisition of plates was undertaken using Image Studio Lite (v 5.2.5) software under the following settings: resolution 42 μm, quality medium and focus 3 mm. The foci were counted in each well and viral titers were calculated and expressed as FFU/mL. RRV iPAs were performed at 12 hpi using the same method and anti-RRV E1 protein monoclonal antibody G8. Viral titers determined by iPAs were expressed in FFU/mL.

**Viral growth kinetics**. Growth kinetics for SARS-CoV-2 were performed on Vero E6 cells and MOI = 0.01. Growth kinetics for RRV were performed on Vero 76 or C6/36 cells at MOI = 0.1. Growth kinetics for CASV were performed on C6/36 cells at MOI = 0.1.

**ZsGreen live fluorescence and flow cytometry of SARS-CoV-2 ZsGreen reporter virus-infected cells**. For live fluorescence, Vero E6 cells were seeded on coverslips in 24-well plates and infected with MOI = 0.1 of SARS-COV-2 ZsGreen virus or WT QLD02 isolate. Coverslips with infected cells were fixed for 30 min at 4 °C with 4% paraformaldehyde (PFA) at 1 and 2 dpi and washed with PBS. The ZsGreen fluorescence was ether visualized directly on Nikon Eclipse Ts2 Epi-fluorescent Inverted Microscope at 40× magnification, and 488 nm filter (Fig. 2g), or coverslips were first mounted on glass slides in ProLong Diamond Antifade Mountant with DAPI (Invitrogen, USA) and then imaged on ZEISS LSM 710 laser scanning confocal microscope at 100× magnification with oil emersion and 488 nm filter for ZsGreen or 405 nm filter for DAPI (Fig. 2h).

For flow cytometry, Vero E6-TMPRSS2 cells were seeded at $1 \times 10^6$ cells/well overnight and infected with SARS-COV-2 ZsGreen2 virus at MOI 0.1. Two days post infection, the supernatant was used to passage the virus two more times on Vero E6-TMPRSS2 using MOI = 0.1. In each passage, supernatants and cells were collected. Supernatants were used for virus quantification by iPA, and cells were used to analyze ZsGreen expression by flow cytometry. The cells were washed with PBS, treated with 500 μL of trypsin for 5 min at 37 °C, spun at 500 g for 5 min, and suspended in 500 μL of DMEM (containing 2% FCS and P/S). Cells were fixed with 4% PFA for 30 min. To remove the PFA, cells were spun down at 500 g for 5 min and suspended in 1 mL of the ice-cold PBS containing 5% heat-inactivated FCS once. Lastly, cells were spun down at 500 g for 5 min and then suspended in ice-cold PBS and passed through 70 μm mesh filter prior to BD Accuri™ C6 analysis. Data were then analyzed using Flowjo v10.

**SARS-CoV-2 mouse infections and determination of viral titers in organs and tissues**. Mouse experiments were approved by the QIMR Berghofer MRI Biosafety Committee and Animal Ethics Committee (project P3600) and conducted in accordance with the "Australian Code for the care and use of animals for scientific purposes" as defined by the National Health and Medical Research Council of Australia. Work was conducted in a biosafety level-3 (PC3) facility at the QIMR

Berghofer MRI (Australian Department of Agriculture, Water and the Environment certification Q2326 and Office of the Gene Technology Regulator certification 3445). Heterozygous K18-hACE2-transgenic mice (B6.Cg-Tg(K18-ACE2)2Prlmn/J, The Jackson Laboratory, Bar Harbor, ME, USA) were bred in-house by crossing with C57BL/6J mice (Animal Resources Center, Canning Vale, WA, Australia). DNA from the tail was isolated using Extract-N-Amp Tissue PCR kit (Sigma) and PCR genotyping undertaken as described (The Jackson Laboratory. Genotyping protocols database, Protocol 37654) except using primers forward—5′-CTTGGTGATATGTGGGGTAGA-3′, reverse 5′ CGCTTCATCTCCCACCACTT-3′ (recommended by NIOBIOHN, Osaka, Japan). Thermocycling conditions were 94 °C 3 min, 35 cycles of 94 °C 30 s, 55.8 °C 30 s, 72 °C 1 min, and final extension of 72 °C 10 min. The conditions the mice were kept are as follows: light = 12:12 h dark/light cycle, 7:45 a.m. sunrise and 7:45 p.m. sunset, 15 min light dark and dark light ramping time. Enclosures: M.I.C.E cage (Animal Care Systems, Colorado, USA). Ventilation: 100% fresh air, eight complete air exchange/h/rooms. In-house enrichment: paper cups (Impact-Australia); tissue paper, cardboard rolls. Bedding: PuraChips (Able scientific) (aspen fine). Food: Double bagged norco rat and mouse pellet (AIRR, Darra, QLD). Water: deionized water acidified with HCl (pH = 3.2). QLD02 and CPER-derived viruses were used to inoculate male 6–8-week old K18-hACE2 mice ($n = 3$ per group) intranasally with $8 \times 10^4$ FFU of each virus in 50 µL medium while under light anesthesia; 3% isoflurane (Piramal Enterprises Ltd., Andhra Pradesh, India) delivered using The Stinger, Rodent Anesthesia System (Advanced Anaesthesia Specialists/Darvall, Gladesville, NSW, Australia). Mice were euthanized at day 5 post infection using $CO_2$, and tissues were homogenized using four ceramic beads at 6000 rpm twice for 15 s (Precellys 24 Homogenizer, Bertin Instruments, Montigny-le-Bretonneux, France). After centrifugation for 10 min, $9400 \times g$ at 4 °C, virus titers in supernatants were determined by $CCID_{50}$ assays using Vero E6 cells.

**Histology and scoring of mouse lung sections**. Lungs were fixed in 10% formalin, embedded in paraffin, and sections stained with H&E (Sigma-Aldrich, Darmstadt, Germany). Slides were scanned using Aperio AT Turbo (Aperio, Vista, CA, USA) and images extracted using Aperio ImageScope software v12.3.2.8013 (Leica Biosystems, Wetzlar, Germany). Automatic quantitation of nuclei count and whitespace of H&E stained sections was undertaken using QuPath v0.2.3[68].

**RRV ELISA-based MXRA8-Fc binding assays**. Purified RRV (50 µL, 10 µg/mL) was coated onto maxi-sorb ELISA plates (Thermo Fisher) overnight in PBS, pH 7.4. Plates were blocked with PBS supplemented with 1X KPL (SeraCare) for 1 h at room temperature. Human MXRA8-Fc (residues 20-337, UniProtKB: Q9BRK3) and negative human PD1-Fc controls were serially diluted in PBS supplemented with 1X KPL, 50 µL/well was added for 1 h at 37 °C. Plates were washed four times with PBS containing 0.05% Tween-20. Plates were then incubated with 50 µL/well horseradish peroxide conjugated goat anti-human IgG (H+L) (Goat anti-Human IgG Secondary Antibody, HRP (Invitrogen, Thermo Fisher, Cat number: A18829, RRID: AB_2535606)) at 5 µg/mL in PBS supplemented with 1X KPL for 1 h at 37 °C. After five washes with PBS containing 0.05% Tween-20, plates were developed with TMB (Life Technologies) and 2 M $H_2SO_4$. Absorbance was read at 450 nm.

**MNV western blot**. Cell lysates from Raw264.7 cells infected with CPER-derived or WT MNVs or Mock were harvested in KALB lysis buffer (150 mM NaCl, 50 mM Tris-HCl, pH 7.5, 1% (v/v) Triton X-100, 1 mM EDTA) supplemented with protease inhibitor cocktail III (Austral Scientific). Cell lysates from Raw264.7 cells infected with CPER-derived or WT MNVs or Mock were harvested in KALB lysis buffer (150 mM NaCl, 50 mM Tris-HCl, pH 7.5, 1% (v/v) Triton X-100, 1 mM EDTA) supplemented with protease inhibitor cocktail III (Austral Scientific). Samples were separated on a polyacrylamide gel and transferred to a polyvinylidene difluoride membrane. Membranes were blocked with 5% BSA/TBS-T for 2 h at room temperature prior to probing with primary antibodies overnight at 4 °C. Primary antibodies, anti-MNV1 capsid antibody 5C4.10 (Cat: MABF2097 Merck Millipore) or anti-actin antibody (Cat: A2066 Sigma), were each diluted 1:1000 in 5% BSA/TBS-T. The following day, the membrane was washed three times with TBS-T and incubated with either anti-mouse HRP secondary antibody (Cat: G-21040, Thermo Fisher Scientific) or anti-rabbit HRP secondary antibody (Cat: A16035, Thermo Fisher Scientific). Secondary antibodies were prepared by 1:10,000 dilution in TBS-T. Probed membranes were developed with the Amersham ECL Western Blotting Detection Reagent and imaged on the GE Healthcare Life Sciences AI600 Imager.

**HuNoV RT-qPCR**. Viral RNA was extracted from the supernatant of HuNoV CPER-transfected NIH3T3 cells using the AccuPrep Viral RNA extraction Kit (Bioneer). cDNA was generated by reverse transcription using Sensifast RT (Bioline) at 25 °C for 10 min, 42 °C for 15 min, and 85 °C for 5 min and quantified using iTaq Universal SYBR Green Supermix (Bio-Rad) under the following cycling conditions: 50 °C for 8 min, 95 °C for 2 min, 40 cycles of 15 s at 95 °C, and 1 min of annealing/extension at 60 °C, followed by a final extension for 10 min. To quantify Viral RNA was extracted from the supernatant of HuNoV CPER-transfected NIH3T3 cells using the AccuPrep Viral RNA extraction Kit (Bioneer). cDNA was generated by reverse transcription using Sensifast RT (Bioline) at 25 °C for 10 min,

42 °C for 15 min, and 85 °C for 5 min. cDNA was quantified using iTaq Universal SYBR Green Supermix (Bio-Rad) under the following cycling conditions: 50 °C for 8 min, 95 °C for 2 min, 40 cycles of 15 s at 95 °C, and 1 min of annealing/extension at 60 °C, followed by a final extension for 10 min. To quantify HuNoV gRNA copies a previously described protocol[69] was used. In brief, a pGEM-T-easy plasmid containing the NS/VP1 overlap region (nucleotides 3521–6916) of HuNoV (GenBank ID: GU445325) gifted by Peter White (University of New South Wales, Australia) was used to generate in vitro transcribed RNA. The RNA was quantified using the Qubit RNA BR assay (Q10211 Thermo Fisher Scientific) and volume adjusted to $10^{10}$ genome copies/µL. A tenfold serial dilution was undertaken to generate a standard curve which was reverse transcribed along with CPER-derived viral RNA and qPCR performed using the NK2PF/NK2PR primer pair[69].

**CASV IFA**. Transfected cell monolayers were removed by adding trypsin and then seeded onto glass coverslips at a density of $1 \times 10^5$ with appropriate supplementation of media and FBS. Once adherent, the supernatant was removed, and the cell monolayers were fixed in 100% ice-cold acetone. Coverslips were blocked with a blocking buffer (0.05 M Tris/HCl (pH 8.0), 1 mM EDTA, 0.15 M NaCl, 0.05% (v/v) Tween-20, 0.2% w/v casein) for 1 h at room temperature. Coverslips were then incubated for 1 h with the mAb 9D7 (anti-CASV) or naive mouse serum (1:1000 in blocking buffer). Following two washes with PBST (PBS with 0.05% Tween-20), coverslips were incubated for 1 h with a goat anti-mouse IgG H+L AlexaFluor 488 (Invitrogen, A-11001) secondary antibody according to the manufacturer's protocols. Immediately after removing the secondary antibody, Hoechst 33342 nuclear stain (Invitrogen, 62249) was used at 1:1000 in the dark for 10 min to stain nuclei. Lastly, the coverslips were subjected to three washes with PBST and then mounted onto glass microscope slides using ProLong Gold Anti-fade (Invitrogen, P36934). All coverslips were imaged with the ZEISS LSM 510 META confocal microscope.

**CASV $TCID_{50}$ assay**. For CASV growth kinetics cell monolayers were inoculated in triplicate with 500 µL of virus (growth medium for the negative control) at an MOI = 0.1. After incubating at room temperature with agitation for 1 h, the inoculum was removed, and the cells were washed three times with PBS before being replenished with 1 mL of 2% FBS in RPMI 1640. Viral supernatant was harvested at 16, 36, and 48 h post infection, and virus titers were determined using $TCID_{50}$ assay on C6/36 cells. For the $TCID_{50}$ assay, supernatant from each time point (16, 36, and 48 h) was titrated onto C6/36 cells at tenfold serial dilutions starting at $10^{-1}$ and ending at $10^{-16}$, with five replicates at each dilution increment. After 48 h, cells were fixed in 20% acetone, 0.02% bovine serum albumin (BSA) in PBS overnight. Titers were confirmed by ELISA with mAb 9D7 (murine anti-CASV spike, School of Chemistry and Molecular Biosciences, UQ)[70].

**Statistical analysis**. For growth kinetics, a two-way ANOVA test with Tukey's multiple comparisons test to compare treatments at each time point was used. For comparisons between nasal and lung titers of SARS-CoV-2 in mice unpaired $t$-test with Welch's correction was used. All data were analyzed using GraphPad Prism software (v9.0.0). The level of statistical significance was set at 95% ($p \leq 0.05$).

**Reporting summary**. Further information on research design is available in the Nature Research Reporting Summary linked to this article.

## Data availability
The consensus sequence for the SARS-CoV-2 QLD02 P3 and P4 cDNA and PCR-amplified fragments used in CPER are available in GenBank with accession number MW772455. Raw sequencing data generated in this study are available in the Sequence Read Archive hosted by the National Center for Biotechnology Information with accession number PRJNA707404. Raw data underlying the results are shown in Figs. 1d, e, 2e, j, m, 3d, g, h, and 4c, f and Supplementary Figs. 2b, 3e, f, and 4b–d are provided in Source Data file. The authors declare that all other data supporting the findings of this study are available within the paper, its supplementary information files or the source data file provided with this paper. Source data are provided with this paper.

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

## Acknowledgements

The authors thank Australian Infectious Diseases Research Centre for a seeding grant to A.A.K. and A.S. for establishing SARS-CoV-2 CPER and a seeding grant to R.A.H., J.H.-P., and A.S. for establishing CASV CPER. A.S. was supported by an Investigator Grant from the National Health and Medical Research Council of Australia (APP1173880). The authors acknowledge funding support from the Medical Research Future Fund (APP1202445-2020 MRFF Novel Coronavirus Vaccine Development Grant to D.W. and P.R.Y.) and Associate Prof. Keith Chappell for project support. The authors thank Clive Berghofer and Lyn Brazil (and others) for their generous philanthropic donations that funded the setup of the PC3 (BSL3) SARS-CoV-2 research facility at QIMR Berghofer MRI. The authors thank Dr I. Anraku for his assistance in managing the PC3 facility at QIMR Berghofer MRI, and Dr Viviana Lutzky for her help with proof reading. Elements of Fig. 1a were generated with BioRender.com. The authors thank Queensland Health, Brisbane, for providing the SARS-CoV-2 virus isolates.

## Author contributions

Conceptualization and experimental design—A.A.K., Y.X.S., J.H.-P., D.W., J.M.M., A.A.A., and N.M.; data acquisition—A.A.K., A.A.A., J.D.J.S., R.P., J.M.D., J.R.P., Y.X.S., D.J.R., T.T.L., N.M., X.W., N.Y.G.P., F.J.T., J.J.H., M.E.F., B.L., C.L.D.M., S.T.M.C., D. J.D.C.G., J.M.H., and D.A.M.; data analysis—A.A.K., Y.X.S., A.A.A., R.P., J.M.D., D.J.R., N.M., M.B., D.A.M., J.M.H., F.C., J.M.M., J.H.-P., A.S., and D.W.; crucial reagents—A.P. and F.M.; project supervision—A.A.K., Y.X.S., F.M., F.C., A.A.A., N.M., R.A.H., P.R.Y., J.M.M., J.H.-P., A.S., and D.W.; writing original draft—A.A.K.; and reviewing and editing —A.A.K., R.P., J.M.D., J.M.M., J.H.-P., R.A.H., D.W., and A.S.

## Competing interests

The authors declare no competing interests.
