## [Peer Review File · Nature Communications]

Reviewers' Comments:

Reviewer #1:

Remarks to the Author:

Remarks to the Editor and Author:

The study by Amarilla et al is a logical development of methodology pioneered by the same group of authors in 2013 (PMID: 23236063) for development of reverse genetics system for positive sense RNA viruses. The main advancement of present work compared to their previous publication is incorporation of polyA tract into 'adapter/linker' fragment which was used for circularization of PCR-amplified viral cDNA fragments. In addition, this work demonstrates that the size of the 'adapter/linker' can be reduced from 4 kb to a relatively short 1 kb fragment. It is debatable whether this represents a major advancement of circular polymerase extension reaction methodology. But in my opinion, given the urgent need for rapid and reliable methods that allow manipulation of SARS-CoV2 genomes, this publication will be of significant interest to laboratories working on this virus worldwide. The manuscript is well written and easy to read, although some sections seem to be redundant and can be shortened/removed. Experiments are well-designed and are appropriate to address the posed questions. Majority of the manuscript's conclusions are supported by experimental data. Statistical analysis of the data is appropriate.

Major comments:

1- I think that authors should clearly state that this work is a continuation of their previous publication (PMID: 23236063) on circular polymerase extension reaction methodology. Authors should address how the present work is different from previous work.

2- In this manuscript authors only discuss advantages of circular polymerase extension reaction as compared to alternative methods for construction of reverse genetics systems. However, authors ignored obvious pitfalls associated with their methods. A- For instance reliance on highly 'transferable' HEK293 or NIH3T3 cells (which normally do not get infected with SARS-CoV2 or HuNoV, respectively) for virus recovery may drive evolution of recovered viral population for adaptation to these particular cell types. This, in turn, may have negative effect on virus fitness in physiologically relevant systems (also see my point in lines 132-138 on physiologically relevant systems below). B - it appears circular polymerase extension method has inherent limitation for how many modifications can be simultaneously introduced into virus genome. Introduction of one additional mutation increases total number of amplified cDNA fragments at least by one additional fragment. Considering that newly characterized variants of SARS-CoV2 (see lineages B.1.1.7, B.1.351, and P.1) contain tens of novel mutations it seems unlikely that all these mutations can be generated in one reaction (this should require perfect assembly of complete 30 kb virus genome from tens of individual PCR reactions). Therefore at least some intermediate plasmid cloning would be required. C - it would be very difficult (or virtually impossible) to introduce repeated sequences into virus genome using PCR extension method since primers can simultaneously anneal to these repeated regions. An example of repeated sequences might be virus cis-acting regulatory regions and/or genome modification which requires introduction of several repeated targets for cellular microRNAs. I believe that all these limitations (A-C) should be discussed in the manuscript.

Minor comments:

Line 13: define word 'efficient'.

Lines 25-30: Picornaviruses are major human pathogens that are also positive sense RNA viruses, which probably should be mentioned here.

Lines 72-75: This repeats part of the introduction and should be shortened/removed.

Line 83: Reference is needed at the end of this sentence.

Lines 103-109: these considerations are obvious and unnecessary. Utilization of HDVr is a standard

practice to generate 3'end of virus genomes.

Lines 112-115 and Fig 1: I believe that it would be important to provide a gel picture of CPER reaction mix in Fig1. This will give readers general understanding of overall efficiency of CPER reaction, informing them what should be expected if they decide to adopt CPER in their research.

Lines 117-119: Please comment whether it would be possible to avoid using HEK293 cells for CPER mix transfection and recover viruses directly in Vero E6 cells (see comment 2A in major comments).

Lines 122-130: Have authors done deep sequencing or Sanger sequencing of WT and CPER 1 and CPER 1 viruses? This information is important to understand genetic integrity and heterogeneity of recovered viruses.

Lines 132-138: authors should discuss limitation of K18-hACE2 mice as they are hypersensitive to SARS-CoV2 infection and do not recapitulate attributes of natural human infection. Have timepoints other than Day 5 been analyzed, and what was the result? I also suggest to show individual data points in Fig. 1e to better represent variability of the nasal and lung titers.

Lines 158: add reference to PMID: 33184236 after reference 18.

Lines 167: I think that data for virus QLD935 is not relevant for this manuscript and can be removed.

Lines 207-208: please check the years of RRV epidemic and provide years of CHIKV epidemic.

Lines 254-276: this section can be removed from the manuscript. Utility of CPER for alphaviruses was sufficiently proven in the previous section. Analysis of E2/E1 positions in interaction with alphavirus receptor is very incomplete and such study requires separate publication.

Lines 279 and 287: repeat of 'positive-strand RNA' should be removed.

Lines 295- 297: please explain why transfection cannot be done directly into RAW264.7 cells.
Line 309: spelling error.

Line 343: refers to a genome size of viruses that has been already provided in the manuscript.

Lines 348- 350: is a repeat of lines 102-109.

Line 354: please define/explain word 'authentic'.

Line 362: please define/explain word 'resemble'.

Lines 393- 396: this statement should be modified according to my comments 2A-2C from section Major comments.

Reviewer #2:

Remarks to the Author:

In this study, a collaborative team lead by the Watterson and Khromykh laboratories describes a novel extension of the circular polymerase extension reaction (CPER) technique to manipulate an impressive panel of positive-stranded RNA viruses that encode poly-adenosine (poly-A) tails. Molecular clone technologies are of critical importance for multiple facets of virology. A variety of techniques exist to create virus stocks with a defined genetic sequence. Continued innovation to simplify these techniques and expand their utility is of broad and significant technical interest.

The experimental quality of the work presented is of the high caliber now expected of these authors. A similar set of experiments is performed for multiple virus types, clearly supporting the overall point of the paper that this technique can be used broadly. While it may be argued the conceptual advance here is modest, this study is a nice roadmap for how to perform this technique using the viruses detailed within and those not included in this comprehensive study. Well done.

An experiment the authors should consider is a more extensive (deeper) sequencing study of SARS-CoV-2 produced by CPER compared to other methods. Sequence heterogeneity is a challenge with the production and propagation of SARS-CoV-2 concerning attenuating mutations at the furin cleave site and other locations. The two-step virus rescue approach detailed here (293T and then Vero) may select for variants of this type (as a minority population) in seed stocks that are expanded during subsequent infections to create working stocks. The variant frequency cannot be estimated using the bulk sequencing procedure used elsewhere in the paper to confirm the presence of the D614G mutation. Have any NGS studies been performed?

Line 483 could be clarified. Are the authors referring to microgram quantities of PCR product, a mass visible on a gel? Additional insight will help those that repeat these techniques.

Reviewer #3:

Remarks to the Author:

The authors describe a reverse genetics platform for positive strand RNA viruses with polyA tail. The platform relies on a previously described method, called CPER, that allow to generate wild-type and recombinant viruses without (i) intermediate cloning step, (ii) full-length genome assembly into vector (plasmid, BAC etc.) and (i) in vitro TNA transcription step. In the present work, this versatile method was applied to a large variety of positive strand RNA viruses with polyA tail, including two viruses from the Nidovirales order, the SARS-COV-2 and the CASV. Generating such viruses with large genomes is always laborious and this alternative approach provides a new option to the scientific community. Finally, the authors demonstrated that this method can be employed to rescue mutant and reporter viruses.

Overall, the study is well done and the manuscript is well written. However, this work has several shortcomings which must be corrected.

Major comments

1) lines 81-83: "These manipulations,, prone the introduction of unwanted mutations". I am afraid that the CPER could also lead to the introduction of undesired mutations into viral genomes. There is a PCR step during the CPER procedure. Even when high-fidelity polymerases are used, PCR amplification can generate a low rate of undesired nucleotide changes. Therefore, PCR-based reverse genetics methods — such as the CPER method — are expected to be associated with artificial viral heterogeneity, including potentially consensus mutations. The exploration of this issue is entirely lacking in this manuscript. In order to clarify this issue, the genome of all the viruses generated using this study must be entirely sequenced and compared with that of the original strains (or that of the infectious clone for MNV).

2) lines 99-110 and 347-350: I do not agree with these sentences. For positive strand RNA viruses with polyA tail, reverse genetics systems (bacterium-free or not) always include at the end of the genome a poly tail as in this study. This is not an "innovative design feature". In some systems under the control of pCMV, a polyA signal (eg SV40 polyadenylation signal), generally incorporated downstream the HDVr sequence, is used as termination signal to increase RNA processing/stability in cellulo (for example: Enterovirus A71 DNA-Launched Infectious Clone as a Robust Reverse Genetic Tool, Tan et al. Plos one 2016). This signal is not use to add the polyA tail at the end of the viral genome and is also present in reverse genetic systems for positive strand RNA viruses without polyA tail (for example: A Novel Approach for the Rapid Mutagenesis and Directed Evolution of the Structural

Genes of West Nile Virus Lin et al. J Virol 2012). Please, clarify the manuscript.

3) line 19 "generating huNoV directly from RNA isolated from a clinical sample", lines 302-312 and lines 402-405. The data presented in the manuscript are insufficient to claim that a huNoV was generated directly from RNA isolated from a clinical sample. "A clear increase in viral RNA was observed between 3hpt and 72hpt" is the only result presented. The virus need to be passaged at least once in cell culture.

Minor comments

-Title: why not add at the end "with polA tail"?

-lines 24-30: I think that few words about picornaviruses, that are major human pathogens, would be useful.

Line 52: the reference nb 6 is the original description of the CPEC method, slightly different of the CPER method. Please add a more recent reference for the CPER method.

-line 71: 2021?

-lines 138-145 and fig 1F-H: Why not use a scoring strategy to compare more precisely lungs from animals infected with CPER-derived or WT viruses? Blind analysis of the samples?

-Paragraph 179-196: Why not assess the stability of the reporter expression over serial passages in cell culture? To do that, you need a quantitative method to assess ZsGreen expression in cell cultures.

Lines 291-300: An infectious clone was used as template to perform PCR amplifications. Did you verify that the plasmid present in the PCR products and finally in CPER products was not able to produce infectious particles? (any control or plasmid removal step?)

-lines 417-424: please provide grow/transfection temperatures

- lines 427-440: please provide, if available, the passage history for each strain.

-Figure 1e: there is a difference between WT and CPER-infected animals. This difference is not significant probably because very few animals were used. Please, discuss this issue in the manuscript.

-Figure 2j, 3d, 3g, and 4j: please add SD in the figures

The reviewers comments have been recapitulated in their entirety in italics.

Reviewer #1 (Remarks to the Author):

Major comments:

1- I think that authors should clearly state that this work is a continuation of their previous publication (PMID: 23236063) on circular polymerase extension reaction methodology. Authors should address how the present work is different from previous work.

This is now clearly stated in the **Introduction** (lines 47-49): “Herein we advanced the circular polymerase extension reaction (CPER) methodology that we previously developed for flaviviruses⁶⁻⁸ to allow the generation of RNA viruses that have large genomes and that contain polyA tails...” We have also added to the **Discussion** how our present work is different from our previous CPER developmements with flaviviruses (lines 378-381): “The ability to recover by CPER RNA viruses that have large genome and that contain polyA tail represent a significant advance from our previous CPER developments for flaviviruses which have relatively small (11kb) RNA genome and do not contain polyA tail.”

We have also provided a detailed description of our CPER methodology as it applies to polyA viruses in the first paragraph of the Discussion.

2- In this manuscript authors only discuss advantages of circular polymerase extension reaction as compared to alternative methods for construction of reverse genetics systems. However, authors ignored obvious pitfalls associated with their methods.

A- For instance reliance on highly ‘transferable’ HEK293 or NIH3T3 cells (which normally do not get infected with SARS-CoV2 or HuNoV, respectively) for virus recovery may drive evolution of recovered viral population for adaptation to these particular cell types. This, in turn, may have negative effect on virus fitness in physiologically relevant systems (also see my point in lines 132-138 on physiologically relevant systems below).

The reviewer is quite correct; for instance, SARS-CoV-2 virus propagation in Vero E6 cells is well known to lead to the selection of mutations in the multibasic S1/S2 furin cleavage site in the spike protein. However, this *in vitro* evolution/selection issue is not restricted to the CPER methodology and also occurs in other SARS-CoV-2 reverse genetics systems (e.g. Thao et al. Nature 2020, Diamond Nat Med, 2021, Chiem et al. J Virol 2021, Rihn et al. PLoS Biology 2021). We have thus added the following sentence to **Developing CPER for SARS-CoV-2** in the Results (lines 106-108); “Culture of SARS-CoV-2 viruses in Vero E6 cells is well known to select rapidly for mutations, often in or around the furin cleavage site¹⁹⁻²². This remains true for a number of reverse genetics systems^{16,21,23}, although deep sequencing of recovered viruses is not always provided^{15,17,24-26}.”

To specifically address this issue for CPER, we have deep sequenced the parental SARS-CoV-2 QLD02 isolate, cDNA fragments used for CPER assembly, and five CPER-generated SARS-CoV-2 viruses (8 deep sequencing data sets). This identified a small number of changes in CPER-recovered viruses introduced during virus recovery/propagation. We have presented the data in a new Supplementary Fig 2 and Supplementary Tables 1 and 2.

As pointed out by the reviewer, HEK293 cells do not get infected with SARS-CoV2, so we also examined the use of ACE2-expressing 293T cells for CPER transfection or the TMPRSS2-expressing Vero E6 cells for virus recovery (CPER3 and ZsGreen2 viruses, respectively, in the new Supplementary Fig 2 and Supplementary Tables 1 and 2). The use of ACE2-expressing HEK293 cells for CPER transfection (CPER3 virus) reduced the number of unintended changes to a single amino acid change, thus achieving nearly complete (99.99%) amino acid sequence fidelity. The use of TMPRSS2-expressing Vero E6 cells for CPER virus recovery and virus passaging (ZsGreen 2 virus) avoided selection of furin site mutations as also reported by others (Sasaki et al PLoS Path 2021, Rhin et al PLoS Biol 2021). These new experiments are described in the Results under the **Developing CPER for SARS-CoV-2** (Lines 95-117), and the **Utilizing CPER for the generation of SARS-CoV-2 D614G mutant and ZsGreen-expressing reporter virus** (Lines 192-197, 204-211) headings.

B – it appears circular polymerase extension method has inherent limitation for how many modifications can be simultaneously introduced into virus genome. Introduction of one additional mutation increases total number of amplified cDNA fragments at least by one additional fragment. Considering that newly characterized variants of SARS-CoV2 (see lineages B.1.1.7, B.1.351, and P.1) contain tens of novel mutations it seems unlikely that all these mutations can be generated in one reaction (this should require perfect assembly of complete 30 kb virus genome from tens of individual PCR reactions). Therefore at least some intermediate plasmid cloning would be required.

The reviewer is considering a scenario where a new variant is sought, but the only virus available to the researcher would require introducing, say 10 changes across the genome. As we have shown in Fig. 2a, to introduce a change in one fragment F5, two fragments were used, F5A and F5B. Thus to introduce 10 separated changes across the genome, 16 fragments would be needed etc. Indeed, we do not know how many fragments CPER can tolerate in such scenarios.

To address the reviewer’s comment, we have rephrased the section in the **Discussion** (Lines 417-429) to read “Conceivably, multiple mutations, deletions and/or insertions can be simultaneously introduced by CPER by swapping multiple WT fragments with corresponding fragments containing the desired mutations/deletions and/or by adding desired insertion fragments. However, we have not explored the upper limit of the number of cDNA fragments that can be successfully assembled using CPER, with each mutation/insertion requiring the addition of another fragment to the CPER reaction. If multiple mutations are needed in one fragment, then this fragment could be generated synthetically and be incorporated into the CPER reaction. We have shown previously that an infectious virus can be generated entirely from synthetic DNA fragments using CPER⁷. If mutations are needed in multiple fragments, a mixture of synthetic and split fragments (e.g. F5a and F5b in Fig. 2a) might be envisaged. Clearly, cDNA fragments derived from one virus isolate can also be used together with cDNA fragments from another virus isolate (and/or synthetic fragments) to generate viruses with a combination of desired mutations⁸.”

C - it would be very difficult (or virtually impossible) to introduce repeated sequences into virus genome using PCR extension method since primers can simultaneously anneal to these repeated regions. An example of repeated sequences might be virus cis-acting regulatory regions and/or genome modification which requires introduction of several repeated targets for cellular microRNAs. I believe that all these limitations (A-C) should be discussed in the manuscript.

We agree with the reviewer that repetitive regions in genomes can complicate the design of CPER assemblies. For instance, SARS-COV-2 already contains a number of TRS repeats, which we avoided in our primer design. To clarify this point, we have added a line to Supplementary Table 3 legend on primer design; “Primers were designed to avoid repeat regions (TRS) (Kim et al., 2020, Cell 181, 914–921)”.

Incorporation of clusters of repeated sequences, for example, miRNA target sites (e.g. PLoS ONE 8(9): e75802) could be achieved by using synthetic fragments. We have thus added to the end of the modified paragraph 3 of the Discussion (lines 429-431); “Synthetic fragments could also, for instance, be used to incorporate clusters of repeated sequences (e.g. microRNA target sites) as long as primers to amplify the fragments are located outside the repeat regions”.

Minor comments:

Line 13: define word ‘efficient’.

Removed

Lines 25-30: Picornaviruses are major human pathogens that are also positive sense RNA viruses, which probably should be mentioned here.

Added (line 21)

Lines 72-75: This repeats part of the introduction and should be shortened/removed.

This text has been deleted to avoid repetition with Introduction, and the Introduction has been considerably rewritten.

Line 83: Reference is needed at the end of this sentence.

As the text is deleted this is no longer required.

Lines 103-109: these considerations are obvious and unnecessary. Utilization of HDVr is a standard practice to generate 3’ end of virus genomes.

Deleted

Lines 112-115 and Fig 1: I believe that it would be important to provide a gel picture of CPER reaction mix in Fig1. This will give readers general understanding of overall efficiency of CPER reaction, informing them what should be expected if they decide to adopt CPER in their research.

We now include a representative gel picture of CPER assembly in Supplementary Fig 1.

Lines 117-119: Please comment whether it would be possible to avoid using HEK293 cells for CPER mix transfection and recover viruses directly in Vero E6 cells (see comment 2A in major comments).

CPER relies on efficient DNA transfection. DNA transfection efficiency of Vero E6 cells is, unfortunately, very poor. Hence transfection is best to be performed in HEK293 cells transfected with DNA with very high efficiency. As HEK293T cells do not support SARS-CoV-2 infection, co-culture of CPER-transfected HEK293T cells with Vero E6 cells is needed to generate productive infection. This is now added to **Discussion** (Lines 442-445).

Lines 122-130: Have authors done deep sequencing or Sanger sequencing of WT and CPER 1 and CPER 1 viruses? This information is important to understand genetic integrity and heterogeneity of recovered viruses.

We now provide deep sequencing data in the new Supplementary Tables 1 and 2 and added their description to the appropriate sections in the text (lines 98-104, 109-117). See also response to comment 2A above and to major comment 1 of reviewer 3.

Lines 132-138: authors should discuss limitation of K18-hACE2 mice as they are hypersensitive to SARS-CoV2 infection and do not recapitulate attributes of natural human infection.

We have added the following sentence to **Developing CPER for SARS-CoV-2** in the Results to address this issue, with some recent publications that nicely summarise this issue (lines 131-135); “ These (K18-hACE2) mice develop a respiratory disease resembling severe COVID-19^{25,26}, but also present with a fulminant brain infection that is associated with mortality²⁷. Although the virus can be found in the brain of \approx 20% of COVID-19 patients, neurological signs and symptoms are thought to arise from systemic reactions or complications rather than being associated with extensive brain infection²⁸.

Have timepoints other than Day 5 been analyzed, and what was the result?

Only day 5 was analyzed.

I also suggest to show individual data points in Fig. 1e to better represent variability of the nasal and lung titers.

Individual data points are now shown

Lines 158: add reference to PMID: 33184236 after reference 18.

This has now been added as requested.

Lines 167: I think that data for virus QLD935 is not relevant for this manuscript and can be removed.

We feel it is important to compare the CPER-generated D614G mutant virus's properties with the clinical isolate QLD935, rather than only comparing CPER-generated D614G mutant virus with QLD02. This is particularly pertinent because the CPER-generated D614G (on a QLD02 background) did not recapitulate the growth properties of QLD935, despite the fact both viruses have the D614G mutation. Thus other mutations present in QLD935 are responsible for the altered growth properties. We have modified the section describing this result to make clearer why we have retained this data (Lines 169-181).

Lines 207-208: please check the years of RRV epidemic and provide years of CHIKV epidemic.

This has been corrected (Lines 215-220).

Lines 254-276: this section can be removed from the manuscript. Utility of CPER for alphaviruses was sufficiently proven in the previous section. Analysis of E2/E1 positions in interaction with alphavirus receptor is very incomplete and such study requires separate publication.

While we recognize that receptor interaction work is limited in scope, we believe it serves as a good example of the power of the approach beyond that described in the previous section as it demonstrates that CPER is compatible with the generation of a panel of whole virion mutants that can be assessed for a specific biological function in a streamlined manner. Further, this work is the first multi-site mutational analysis of alphavirus-receptor interactions performed in a native virion context, and the only study to investigate E1 site-specific receptor linked mutations. Therefore, these findings can be considered a complementary extension of

previously published work that used a recombinant E2 library to probe chikungunya virus and MXRA8 interactions. While these findings may be incremental, they are important as they provide proof-of-principle for CPER enabled whole virus particle receptor interaction studies.

Lines 279 and 287: repeat of 'positive-strand RNA' should be removed.

This has been corrected.

Lines 295- 297: please explain why transfection cannot be done directly into RAW264.7 cells.

The following has been added to the text (lines 312-315): “As NIH3T3 cells are more efficiently transfected with DNA than RAW264.7 cells, CPER was first transfected into NIH3T3 cells...”

Line 309: spelling error.

Corrected

Line 343: refers to a genome size of viruses that has been already provided in the manuscript.

Corrected

Lines 348- 350: is a repeat of lines 102-109.

This has been rephrased

Line 354: please define/explain word 'authentic'.

We mean “viruses with replicative and pathogenic characteristics (for SARS-CoV-2) indistinguishable from wild-type viruses”. We have rephrased the previous sentence to define more clearly our meaning of “authentic”.

Line 362: please define/explain word 'resemble'.

We have modified ‘resemble’ to “in some ways similar to”

Lines 393- 396: this statement should be modified according to my comments 2A-2C from section Major comments.

Modified – see responses to comment 2

Reviewer #2 (Remarks to the Author):

An experiment the authors should consider is a more extensive (deeper) sequencing study of SARS-CoV-2 produced by CPER compared to other methods. Sequence heterogeneity is a challenge with the production and propagation of SARS-CoV-2 concerning attenuating mutations at the furin cleave site and other locations. The two-step virus rescue approach detailed here (293T and then Vero) may select for variants of this type (as a minority population) in seed stocks that are expanded during subsequent infections to create working stocks. The variant frequency cannot be estimated using the bulk sequencing procedure used elsewhere in the paper to confirm the presence of the D614G mutation. Have any NGS studies been performed?

We now provide deep sequencing data for the CPER-generated SARS-CoV-2 viruses (presented in new Supplementary Fig 2 and Supplementary Tables 1 and 2) and their description in the corresponding sections in the text. Specifically the following has been added to **Developing CPER for SARS-CoV-2** (lines 106-117): “Culture of SARS-CoV-2 viruses in Vero E6 cells is well known to select rapidly for mutations, often in or around the furin

cleavage site¹⁸⁻²¹. This remains true for a number of reverse genetics systems^{16,20,22}. Such selection was also seen for CPER viruses recovered from Vero E6 cells (co-cultured with transfected HEK293T) (Supplementary Table 1). Selection of certain amino acid changes was different for CPER1, 2 and 3 viruses (Supplementary Table 1), with such variability in selection also reported for other reverse genetics systems^{16,20,24}. As noted previously^{20,21}, the choice of cell lines can effect these selection processes, with our recovery of CPER3 virus using ACE2-HEK293T cells showed a decrease in the number of affected sites from 4 to 1 (Supplementary Table 1). Given the SARS-CoV-2 genome encodes >9700 amino acids, the CPER method using ACE2-HEK293T cell thus achieved nearly complete (99.99%) amino acid sequence fidelity.”

In addition, Vero TMPRSS2-expressing cells were used for recovery of CPER-generated ZsGreen2 virus and this avoided selection of furin cleavage site mutations. The following text has now been added to **Utilizing CPER for the generation of SARS-CoV-2 D614G mutant and ZsGreen-expressing reporter virus** in Results (lines 192-197): “Additionally, TMPRSS2-expressing Vero E6 cells were generated (Supplementary Note 1) and used in a separate experiment to co-culture CPER-transfected HEK293T cells to generate ZsGreen2 virus (Supplementary Fig 2). Nanopore sequencing of the reporter viruses showed significantly less variation in the furin cleavage site than in CPER1-3 viruses, with the least variation observed in the ZsGreen2 virus generated in TMPRSS2-Vero E6 cells (Supplementary Table 2).”

See also response to comment 2A of reviewer 1 and to comment 1 of reviewer 3.

Line 483 could be clarified. Are the authors referring to microgram quantities of PCR product, a mass visible on a gel? Additional insight will help those that repeat these techniques.

This has now been moved to the Methods and clarified. No, we did not use a mass calculation derived from a gel, we used 0.1 pM, determined based on DNA concentration measured by Nanodrop and the size of the DNA fragment.

Reviewer #3 (Remarks to the Author):

Major comments

1) lines 81-83: “These manipulations, …, prone the introduction of unwanted mutations”. I am afraid that the CPER could also lead to the introduction of undesired mutations into viral genomes. There is a PCR step during the CPER procedure. Even when high-fidelity polymerases are used, PCR amplification can generate a low rate of undesired nucleotide changes. Therefore, PCR-based reverse genetics methods — such as the CPER method — are expected to be associated with artificial viral heterogeneity, including potentially consensus mutations. The exploration of this issue is entirely lacking in this manuscript. In order to clarify this issue, the genome of all the viruses generated using this study must be entirely sequenced and compared with that of the original strains (or that of the infectious clone for MNV).

We now provide deep sequencing data showing that CPER fragments generated by PCR from the SARS-CoV-2 cDNA have 100% identical sequence to the viral cDNA (new Supplementary Table 1), illustrating the high fidelity of PCR reaction (perhaps to be expected from GXL DNA polymerase). This has now been added to the **Developing CPER for SARS-CoV-2** in the Results (lines 98-104): “Nanopore sequencing of the viral cDNA fragments

amplified from P4 viral cDNA showed they had essentially the same swarm/quasi-species sequence variation as the WT passage 4 (P4) virus (Supplementary Table 1). Comparison of the complete consensus nucleotide sequences (Gen bank ID: MW772455) also illustrated the PCR amplification had not introduced any significant changes. These data illustrate that the high-fidelity PCR used to generate the cDNA fragments that were subsequently used for the CPER reactions, had faithfully amplified the original viral sequences.”

We also now provide deep sequencing data for five SARS-CoV-2 viruses generated by CPER, which show an overall very low number (one to four) of amino acid changes, including a change in the furin cleavage site in Spike protein (new Supplementary Table 1). This is commonly found in SARS-CoV-2 viruses propagated in Vero E6 cells and also reported for other reverse genetics systems (see the response to reviewer 2). We also now show that the use of cDNA from lower passage SARS-CoV-2 virus (passage 3 instead of passage 4) and TMPRSS2-expressing Vero E6 (co-cultured with transfected HEK293 cells) retained authentic furin cleavage site (ZsGreen2 virus, new Supplementary Fig 2 and Supplementary Table 2). Additionally, we now show that CPER transfection into ACE2-expressing HEK293 cells reduced the number of amino acid changes from 4 to 1 (CPER3 virus, new Supplementary Fig 2 and Supplementary Table 1). See also response to reviewer 2.

It is worth noting that for currently published SARS-CoV-2 reverse genetics systems, deep sequencing of recovered/recombinant viruses have either not been undertaken (Xie et al. Cell Host Microbe 2020, Xie et al. Nat Protocols 2021, Hou et al. Cell 2020, Hou et al. Science 2020) or raw deep sequencing data are not available for download (Ye et al. mBio 2020, Rihn et al. PLoS Biology 2021). Of the SARS-CoV-2 reverse genetics systems that have presented deep sequencing data of rescued viruses, many show some unintentional non-synonymous changes (Thao et al. Nature 2020 [SRA BioProject: PRJNA615319], Chen et al., Nat Med, 2021 (SRA: PRJNA698378, Supplementary Table 1 of that Manuscript), Chiem et al. J Virol 2021 [SRA Bioproject: PRJNA678001] or a large proportion of reads indicating deletions in the furin cleavage site (Thao et al. Nature 2020 [SRA SRR11426419]).

While deep sequencing of all viruses generated by CPER could be a useful addition to the manuscript, we believe that providing extensive deep sequencing analysis of five CPER-generated SARS-CoV-2 viruses with the largest genome size, together with our previous CPER publications on flaviviruses is sufficient to demonstrate the fidelity of CPER methodology.

2) lines 99-110 and 347-350: I do not agree with these sentences. For positive strand RNA viruses with polyA tail, reverse genetics systems (bacterium-free or not) always include at the end of the genome a poly tail as in this study. This is not an “innovative design feature”. In some systems under the control of pCMV, a polyA signal (eg SV40 polyadenylation signal), generally incorporated downstream the HDVr sequence, is used as termination signal to increase RNA processing/stability in cellulo (for example: Enterovirus A71 DNA-Launched Infectious Clone as a Robust Reverse Genetic Tool, Tan et al. Plos one 2016). This signal is not use to add the polyA tail at the end of the viral genome and is also present in reverse genetic systems for positive strand RNA viruses without polyA tail (for example: A Novel Approach for the Rapid Mutagenesis and Directed Evolution of the Structural Genes of West Nile Virus Lin et al. J Virol 2012). Please, clarify the manuscript.

The reviewer correctly points out that the additions of polyA tail prior to HDVr to generate RNA with polyA tail and of SV40 polyA signal downstream of the HDVr as transcription termination signal was indeed used in CMV-based infectious clones before (e.g. Tan et al. PLOS ONE 2016). Hence, we removed the words “innovative design feature” and

corresponding text in the Results and Discussion and added the Tan et al. PLoS ONE 2016 reference. Also, our flavivirus linker fragment that was used for generating all other linker fragments in this study does contain SV40 polyA signal downstream of HDVr for transcription termination (Setoh et al., JGV 2015). Although this linker was referenced in the Supplementary Note 2, we agree this design element was inadequately announced in the original submission and has now been added in Fig 1a, Results (line 82), Discussion (lines 366-367) and Supplementary Note 2.

3) line 19 “generating huNoV directly from RNA isolated from a clinical sample”, lines 302-312 and lines 402-405. The data presented in the manuscript are insufficient to claim that a huNoV was generated directly from RNA isolated from a clinical sample. “A clear increase in viral RNA was observed between 3hpt and 72hpt” is the only result presented. The virus need to be passaged at least once in cell culture.

We have changed the wording from “illustrating productive infection” to “indicating virus assembly and release” (lines 326-327). HuNoV does not infect any available cell lines – virus propagation (cell to cell infection) requires enteric organoids or zebrafish larvae, systems to which we do not have access. We believe that ~1000-fold increase in the viral RNA copies in the supernatants of transfected cells from 3h to 72h post-transfection provides good evidence of virus production and release and have added more details about the qPCR assay (lines 752-757).

Minor comments

-Title: why not add at the end “with polA tail”?

Although possible, the CPER methodology can be applied to viruses with and without polyA tails.

-lines 24-30: I think that few words about picornaviruses, that are major human pathogens, would be useful.

This has been added (line 21).

Line 52: the reference nb 6 is the original description of the CPEC method, slightly different of the CPER method. Please add a more recent reference for the CPER method.

Additional references for CPER method have been added (line 48).

-line 71: 2021?

This has been corrected.

-lines 138-145 and fig 1F-H: Why not use a scoring strategy to compare more precisely lungs from animals infected with CPER-derived or WT viruses? Blind analysis of the samples?

We now provide a computerised automated scoring (using QuPath v0.2.3) of (i) lung consolidation, measured by white space (i.e. loss of air space in alveolar and bronchi), and (ii) cellular infiltration, measured by nuclei count, in hematoxylin and eosin-stained lung sections from JAXK18 hACE2-transgenic mice infected with wild-type (WT) SARS-CoV-2 and CPER-generated SARS-CoV-2 viruses. Two sections of the lung were averaged to give one value per mouse in WT infected (n=3), CPER infected (n=6), and with mock-infected lungs (n=2) as indicative values. These data show no significant differences in both features between WT and CPER viruses are now included in Supplementary figure 2.

-Paragraph 179-196: Why not assess the stability of the reporter expression over serial

passages in cell culture? To do that, you need a quantitative method to assess ZsGreen expression in cell cultures.

We now provide data on the stability of ZsGreen insertion, retention of authentic furin cleavage site, and ZsGreen expression over 3 viral passages in Tmprss2-Vero E6 cells (new Fig 2k-m, Supplementary Fig 4a-d).

Lines 291-300: An infectious clone was used as template to perform PCR amplifications. Did you verify that the plasmid present in the PCR products and finally in CPER products was not able to produce infectious particles? (any control or plasmid removal step?)

pSPORT-T7-MNV1 infectious clone plasmid does not contain mammalian expression promoter. Instead, it contains bacteriophage T7 promoter and MNV RNA needs to be transcribed first in vitro by T7 RNA polymerase and then transfected into cells for virus recovery. Hence the MNV virus cannot be recovered from this plasmid DNA in transfected mammalian cells. We have now added “under control of T7 promoter” to the description of the plasmid in the text (line 309).

-lines 417-424: please provide grow/transfection temperatures

Now provided (lines 467-469)

- lines 427-440: please provide, if available, the passage history for each strain.

Now provided where known (lines 485-497)

-Figure 1e: there is a difference between WT and CPER-infected animals. This difference is not significant probably because very few animals were used. Please, discuss this issue in the manuscript.

We agree with the reviewer that the number of animals was indeed low, however, we believe that collectively, viral titre data and lung pathology data that showed no significant difference between WT and CPER-derived viruses provide an indication that viral properties in mice are similar. The following sentence has been added to the **Developing CPER for SARS-CoV-2** in Results to reflect this (lines 150-152): “While the number of animals was small, collectively the data on viral titres in nasal turbinate and lungs, and pathological changes in lungs indicate that CPER-derived and parental WT QLD02 viruses behave similarly *in vivo*.”

-Figure 2j, 3d, 3g, and 4j: please add SD in the figures

This has been added to all figures and indicated in the figure legends.

Reviewers' Comments:

Reviewer #1:

Remarks to the Author:

Authors adequately addressed my comments and concerns. I do not have further comments.

Reviewer #2:

Remarks to the Author:

I thought this was a solid paper the first time I read it. I do believe it has been improved by the thoughtful comments of the other reviewers. I have no suggestions that would limit this excellent technical study from moving forward.

Reviewer #3:

Remarks to the Author:

The authors provide a improved version of the manuscript that answer almost all the issues raised during the first round of reviewing.

Only one comment regarding sequencing data provided (supplementary table 1 and lines 98-104):

While it is now clear that the method allows to essentially recapitulate the mutant spectrum observed at the WT passage 4, the following sentence is unclear : "Comparison of the complete consensus nucleotide sequences (Genbank ID: MW772455) also illustrated the PCR amplification had not introduced any significant changes.". Did you find consensus mutations? If yes, please provide a new table, with these mutations and their characteristics (position, gene, synonymous/nonsynonymous etc.). I don't think that detection of such mutations compromise the quality of this study, but it must be clearly presented. I am fully agree with the authors, many already published works, some of them in highly prestigious journals, do not consider this issue. This raises a question regarding the quality of such "fast-track" reviewed papers.

REVIEWERS' COMMENTS

Reviewer #3 (Remarks to the Author):

Only one comment regarding sequencing data provided (supplementary table 1 and lines 98-104): While it is now clear that the method allows to essentially recapitulate the mutant spectrum observed at the WT passage 4, the following sentence is unclear : “Comparison of the complete consensus nucleotide sequences (Genbank ID: MW772455) also illustrated the PCR amplification had not introduced any significant changes.”. Did you find consensus mutations? If yes, please provide a new table, with these mutations and their characteristics (position, gene, synonymous/nonsynonymous etc.). I don't think that detection of such mutations compromise the quality of this study, but it must be clearly presented. I am fully agree with the authors, many already published works, some of them in highly prestigious journals, do not consider this issue. This raises a question regarding the quality of such “fast-track” reviewed papers.

Our Response:

We apologise for the miscommunication here and appreciate that the word “significant” is awkward in this context. As we did not find ANY consensus changes between P4 viral cDNA and PCR fragments used in CPER we have removed the word “significant” and rephrased the text to clarify this. As we did not find any consensus changes, we do not feel it is necessary to include an additional table.

This (lines 99-103) now reads as:

“Comparison of the consensus genome sequences between P4 viral cDNA and PCR-amplified fragments used in CPER assembly (deposited to Genbank, ID: MW772455) did not identify any changes. These data illustrate that the high-fidelity PCR used to generate the cDNA fragments that were subsequently used for the CPER reactions, had faithfully amplified the original viral sequence.“